# Integration of visual and antennal mechanosensory feedback during head stabilization in hawkmoths

**Payel Chatterjee, Agnish Dev Prusty, Umesh Mohan, Sanjay P Sane\***

National Centre for Biological Sciences, Tata Institute of Fundamental Research, Bangalore, India

**\*For correspondence:**
sane@ncbs.res.in

**Competing interest:** The authors declare that no competing interests exist.

**Abstract** During flight maneuvers, insects exhibit compensatory head movements which are essential for stabilizing the visual field on their retina, reducing motion blur, and supporting visual self-motion estimation. In Diptera, such head movements are mediated via visual feedback from their compound eyes that detect retinal slip, as well as rapid mechanosensory feedback from their halteres – the modified hindwings that sense the angular rates of body rotations. Because non-Dipteran insects lack halteres, it is not known if mechanosensory feedback about body rotations plays any role in their head stabilization response. Diverse non-Dipteran insects are known to rely on visual and antennal mechanosensory feedback for flight control. In hawkmoths, for instance, reduction of antennal mechanosensory feedback severely compromises their ability to control flight. Similarly, when the head movements of freely flying moths are restricted, their flight ability is also severely impaired. The role of compensatory head movements as well as multimodal feedback in insect flight raises an interesting question: in insects that lack halteres, what sensory cues are required for head stabilization? Here, we show that in the nocturnal hawkmoth *Daphnis nerii,* compensatory head movements are mediated by combined visual and antennal mechanosensory feedback. We subjected tethered moths to open-loop body roll rotations under different lighting conditions, and measured their ability to maintain head angle in the presence or absence of antennal mechanosensory feedback. Our study suggests that head stabilization in moths is mediated primarily by visual feedback during roll movements at lower frequencies, whereas antennal mechanosensory feedback is required when roll occurs at higher frequency. These findings are consistent with the hypothesis that control of head angle results from a multimodal feedback loop that integrates both visual and antennal mechanosensory feedback, albeit at different latencies. At adequate light levels, visual feedback is sufficient for head stabilization primarily at low frequencies of body roll. However, under dark conditions, antennal mechanosensory feedback is essential for the control of head movements at high frequencies of body roll.

## Editor's evaluation

This paper will be of interest to neuroscientists who study navigation and multisensory integration. In it, the authors use several manipulations to convincingly show that hawkmoths use mechanosensory feedback from their antennae to stabilize their head when their body rotates quickly or when they have little visual input. The results are consistent with the hypothesis that control of head angle in insects that lack halteres results from a multimodal feedback loop that integrates visual and antennal mechanosensory feedback. This advances our understanding of how such stabilizing reflexes work beyond Dipteran flies, where much prior work has focused.

## Introduction

For the control of locomotor behaviors such as flight or walking, the ability to stabilize gaze is of central importance for reducing motion blur (*Land, 1999*). In diverse animals, gaze stabilization is achieved through sensorimotor circuits that rapidly sense and correct any deviation from the intended course, typically by eliciting reflexive motor responses of their head and body (*Hardcastle and Krapp, 2016*; *Land, 1999*; *Waespe and Henn, 1987*). Of these, a class of responses called *optomotor reflexes* refer to those behaviors that occur in response to optic flow (*Borst, 2014*). Although gaze stabilization typically requires eye as well as neck movements, in most flying animals including insects (*Hardcastle and Krapp, 2016*; *Kemppainen et al., 2022*; *Land, 1999*), birds (*Hardcastle and Krapp, 2016*; *Necker, 2007*), and bats (*Eitan et al., 2019*), eye movements relative to the head are either reduced or absent. Such animals stabilize their gaze primarily using head and body movements guided by optomotor reflexes.

In insects, optomotor reflexes have mostly been studied in tethered flies (Diptera) using assays in which their heads were fixed to the thorax with glue while they were presented with a moving display (*Borst, 2014*). In such preparations, the entire image stabilization is achieved through compensatory wing movements which may be tracked by infrared sensors or through high-speed videography (e.g. *Duistermars et al., 2007*). The stroke amplitude difference between the wings can then be used to close the loop between display image and wing movement, allowing the fly to control the projection of the visual environment onto their eyes. This approach has been very successful in helping to uncover diverse visual capabilities of insects. However, because the head was glued to the thorax in these preparations, the role of head movements in image stabilization typically fell out of the purview of such studies. Recent work on the fruit flies *Drosophila melanogaster* suggests that head-fixation adversely affects mechanical power output for flight (*Cellini and Mongeau, 2020*). Moreover, head-fixation in tethered *Drosophila* impairs their ability to fixate on specific objects moving relative to ground; if an object is moving relative to a moving ground, wing movements depend on both object and ground motion, but head movements depend only on ground motion (*Fox and Frye, 2014*). Head-fixation has also been shown to affect steering maneuvers in locusts *Locusta migratoria* (*Robert and Rowell, 1992*). Head and wing movements are thus crucial in steering maneuvers during flight and contribute strongly to gaze stabilization.

In Diptera, head stabilization is mediated by feedback from multiple sensory systems. These include visual feedback from their compound eyes and the mechanosensory feedback and/or feedforward control from their halteres which are modified hindwings that detect aerial rotations (*Hengstenberg, 1988*; *Hengstenberg, 1993*; *Schwyn et al., 2011*). Additionally, mechanosensory proprioceptive structures in the ventral neck-prothorax region called prosternal organs also aid in head positioning (*Paulk and Gilbert, 2006*; *Preuss and Hengstenberg, 1992*). The relative contributions of these sensory systems depend on the angular speed of the maneuver because vision typically provides feedback at slower rates than mechanosensors (*Hengstenberg, 1993*). Importantly, during aerial maneuvers, haltere feedback combines non-linearly with visual feedback in some neck motor neurons to determine the compensatory head movements (*Huston and Krapp, 2009*). Thus, this system offers an interesting case study for how multisensory feedback from visual and mechanosensory modalities is integrated during locomotion. Because mechanosensory feedback from halteres is transduced at faster rates than visual feedback, it is hypothesized to play a key role in guiding wing movements during rapid maneuvers, whereas visual feedback mediates slower turns (*Sherman and Dickinson, 2003*).

Fewer studies have focused on head stabilization in freely flying insects as compared to tethered insects. Free-flight studies in diverse insects (honeybees *Apis mellifera*; *Boeddeker and Hemmi, 2010*; *Wehner and Flatt, 1977*, blowflies *Calliphora vicina*; *Hateren and Schilstra, 1999*) show that rapid head turns precede sharp saccadic turns, whereas gaze stabilization between saccades ensures that the visual field largely translates over the retina, thereby enabling depth perception. When flying through a patterned, oscillating drum surrounding their nest entrance, freely flying bees adjust their head angle based on wide-field image movement in their visual field (*Boeddeker and Hemmi, 2010*; *Wehner and Flatt, 1977*). Moreover, visually induced head roll response is limited beyond specific roll frequencies, suggesting constraints to visually mediated gaze stabilization (*Boeddeker and Hemmi, 2010*).

These studies demonstrate that optic flow plays a key role in head stabilization behavior in both tethered and freely flying insects. However, the role of mechanosensory feedback in gaze stabilization has been largely ignored in insects that lack halteres. Is head roll in these insects primarily visually driven or does it also require mechanosensory feedback from some haltere analogue? Head stabilization in wasps appears to be primarily visually mediated, although some degree of feedforward control has been suggested (*Viollet and Zeil, 2013*). Similarly, in day-flying insects such as dragonflies, head stabilization appears to be dominated by visual feedback, and these insects contain adaptations that ensure passive inertial head stabilization (*Gorb, 1999*; *Hardcastle and Krapp, 2016*; *Mittelstaedt, 1950*). Yet, being predatory insects, they achieve precise visual fixation of targets during prey capture in flight (*Lin and Leonardo, 2017*; *Olberg et al., 2007*). The role of mechanosensory feedback in head movements in dragonflies remains an open question, especially as their antennae are very small and they do not possess halteres (*Gewecke et al., 1974*). In contrast, nocturnal ants can stabilize their heads in darkness during walking, suggesting mechanosensory input for stabilization (*Raderschall et al., 2016*). However, the mechanosensors mediating head stabilization in non-Dipteran insects are yet to be identified.

One potential candidate for the mechanosensory mediation of head stabilization is the Johnston's organ, which is a highly sensitive mechanosensory structure situated in the antennal pedicel-flagellar joint of all insects (e.g. *Gewecke, 1974*; *Sant and Sane, 2018*). In hawkmoths, the Johnston's organ consists of hundreds of concentrically arranged scolopidial units that are range-fractioned, which means that the scolopidial units are tuned to different frequency ranges with high sensitivity (*Dieudonné et al., 2014*). Antennal mechanosensors contribute in sensing airflow (*Roy Khurana and Sane, 2016*; *Natesan et al., 2019*; *Taylor and Krapp, 2007*), and maintaining headwind orientation (*Fuller et al., 2014*). In moths, flight becomes unstable when the mechanosensory load on the Johnston's organ is reduced by clipping off their flagella (*Dahake et al., 2018*; *Sane et al., 2007*). In walking crickets, there is evidence that antennae contribute to compensatory head roll (*Horn and Bischof, 1983*) and electrophysiological recordings from the neck motor system in flies further revealed that the ventral cervical nerve motor neuron in flies receive input from the antennae (*Haag et al., 2010*).

To determine the relative contributions of visual and mechanosensory inputs in gaze stabilization during flight, we studied head movements in the nocturnal hawkmoth, *Daphnis nerii*. In our experiments, moths were tethered to rotatable motor shafts, and we measured the compensatory counter-rotation of their head at two different angular rotations of their body under variable light conditions (*Figure 1A–F*; see Materials and methods). Additionally, we assessed their ability to use antennal mechanosensory feedback by ablating and reattaching their antennal flagellum. Our data show that at lower turning rates of the body, visual feedback is essential for head stabilization. However, at higher angular velocities, feedback from the Johnston's organs plays a crucial role in the control of head rotations, consistent with the hypothesis that this feedback combines with vision to modulate compensatory head movements. We also show that head movements are essential for controlled flight. Thus, the observed reduction in flight performance due to the loss of mechanosensory feedback from the antenna may be related to impaired gaze stabilization.

## Results

Head stabilization performance was characterized using three parameters: gain and phase of the response, and compensation error which is a metric that combines both gain and phase. The head-compensation error parameter was computed as the distance between the perfect head stabilization point and the experimental point from a trial (*Figure 1E*; for details, see Materials and methods, also see *Figure 2—figure supplement 2*). Note that all three parameters must be considered to fully interpret a compensatory head response. When the gain is zero, the system offers no response and hence the phase values at this gain are uninterpretable. For non-zero gain, the phase values indicate how closely the response matches the stimulus in phase. However, in many cases, heavily delayed responses may appear as leading the stimulus in phase. Also, because gain and phase are inherently linked, they cannot be interpreted or statistically compared separately. The compensation error is a metric that enables us to statistically analyze these results.

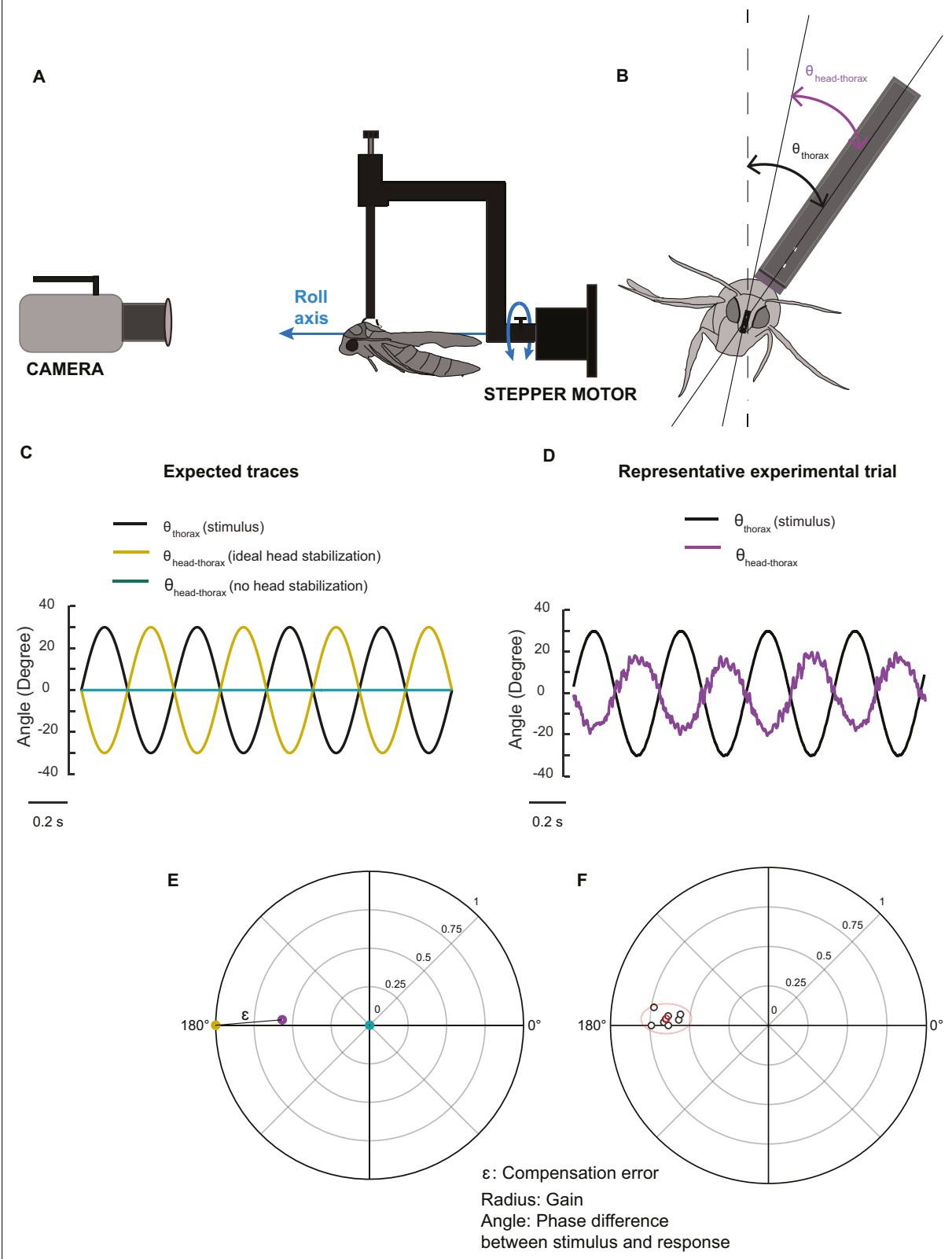

**Figure 1.** Methods and representation of experiments. (**A**) A schematic of the experimental setup illustrating the relative positions of the stepper motor, tethered moth, and camera. (**B**) Front view of the moth with lines drawn through the digitized points and the angles used for analysis. $\theta_{thorax}$ is the angle between the tether position and the vertical and $\theta_{head\text{-}thorax}$ represents angle between head and thorax. (**C**) Traces representing ideal head stabilization ($\theta_{head\text{-}thorax}$: yellow) and no head stabilization ($\theta_{head\text{-}thorax}$: cyan) in response to an imposed roll stimulus of amplitude ±30° and frequency of

*Figure 1 continued on next page*

Figure 1 continued

2 Hz ($\theta_{thorax}$: black). (**D**) Representative experimental trial showing head roll rotations that do not fully compensate the forced body rotations in moths (violet trace, $\theta_{head-thorax}$). (**E**) Polar plot representing gain and phase of the different conditions in (**C**) and (**D**). Each point represents gain in the radial axis and phase in the circular axis. The points are color-coded with reference to the plots in (**C**) and (**D**). Compensation error (labelled ε in the figure, also see Supplementary methods) is the distance between a representative point (purple) and the ideal head stabilization (yellow). The case of zero gain (blue) indicates the absence of any response. (**F**) A cluster of points belonging to a sample treatment is enclosed by a 95% confidence ellipse and the mean of the points is highlighted in red.

The online version of this article includes the following figure supplement(s) for figure 1:

**Figure supplement 1.** The visual environment of the tethered moth.

## Visual cues affect head stabilization at low frequency

To assess the role of visual cues in mediating compensatory head rotations, we provided rotating stimuli to tethered moths at different light levels (Twilight condition: ~250 lux; dark condition: <0.01 lux; see Materials and methods). At each light level, the moths were rotated by ±30° about their longitudinal roll axis at oscillation frequencies of 2 Hz (*Figure 2A*) and 6 Hz (*Figure 2B*), while we measured their compensatory head movements.

Under twilight condition at 2 Hz, the compensation error was low with a median ± standard error of mean of 0.4±0.02 (open circles, *Figure 2A and C*). However, when we rotated the moths at 2 Hz under dark condition (filled circles, *Figure 2A and C*, *Figure 3—figure supplement 1A,C*), a reduction in gain (twilight condition median: 0.6, dark condition median: 0.4) and a shift in phase (twilight condition circular median: 177.6°, dark condition circular median: 302.4°, *Figure 3—figure supplement 3*: Cases 1 and 2) increased the compensation error to 1.2±0.09, suggesting a severe reduction in their ability to generate compensatory head movements (p=0.0078, *Supplementary file 1c*, 1, *Figure 3—figure supplement 1A,C*). Additionally, the compensation error values of moths under dark conditions were more variable, which is also evident in the distribution of points in the polar scatter plots (compare open and filled circles, *Figure 2A*; see also *Figure 2C*, *Supplementary file 1a*). A different method of phase computation (see Materials and methods) yielded a nearly identical result (*Figure 2—figure supplement 1*, compensation error under twilight condition: 0.4±0.02, dark condition: 1.2±0.09, p=0.0078, Wilcoxon signed-rank test). Thus, visual input plays a crucial role in mediating head stabilization at 2 Hz (*Video 1*).

We next increased the oscillation frequency to 6 Hz, and observed that under the twilight condition, the compensation error was 0.7±0.02 (open circles, *Figure 2B–C*). At 6 Hz, the phase-shift between the twilight and the dark condition was reduced in comparison to the 2 Hz case (twilight circular median: 173.7°, dark circular median: 214.2°) whereas the gain between the two conditions was similar (*Figure 2B*, *Figure 3—figure supplement 1B,D*, see also *Figure 3—figure supplement 3* Cases 1 and 2). The compensation error (*Figure 2C*) remained largely unaltered with a median ± standard error of mean of 0.7±0.05 (p=0.5469, *Supplementary file 1c*, 2). However, similar to the 2 Hz case, compensation error was more variable under dark conditions (compare open and filled circles, *Figure 2B*, *Supplementary file 1a*). We got similar results using Fourier-based phase computation. (*Figure 2—figure supplement 1B,C*, compensation error under twilight condition: 0.6±0.02, dark condition: 0.7±0.06, p=0.4609, Wilcoxon signed-rank test). Thus, at 6 Hz, severe reduction of visual feedback did not greatly hamper head stabilization, suggesting that at roll rotation of higher frequency, an additional – perhaps mechanosensory – feedback played a role in head stabilization. However, the phase-shift and increased variability in the dark condition response suggests that some visual feedback may be required even at 6 Hz.

## Antennal inputs affect head stabilization primarily at high frequency

Mechanosensory feedback from the antennae has been previously shown to mediate flight control in hawkmoths (*Dahake et al., 2018*; *Sane et al., 2007*). To test the hypothesis that it also plays a role in head stabilization, we clipped the antennal flagella, thus greatly reducing the inertial load of the antenna. We subjected these moths to roll stimuli (±30°) at oscillation frequencies of 2 Hz and 6 Hz in both twilight and dark conditions, and filmed the resulting head movements (*Figure 3A–E*).

Under both twilight and dark conditions at 2 Hz (*Figure 3E*), the compensation error was significantly different between the *flagella-intact* (twilight: 0.4±0.02; dark: 1.2±0.09) and *flagella-clipped* groups (twilight: 0.5±0.01; dark: 1±0.01, p=0.0107, p=0.0133, *Supplementary file 1c*, 3, 4). Under

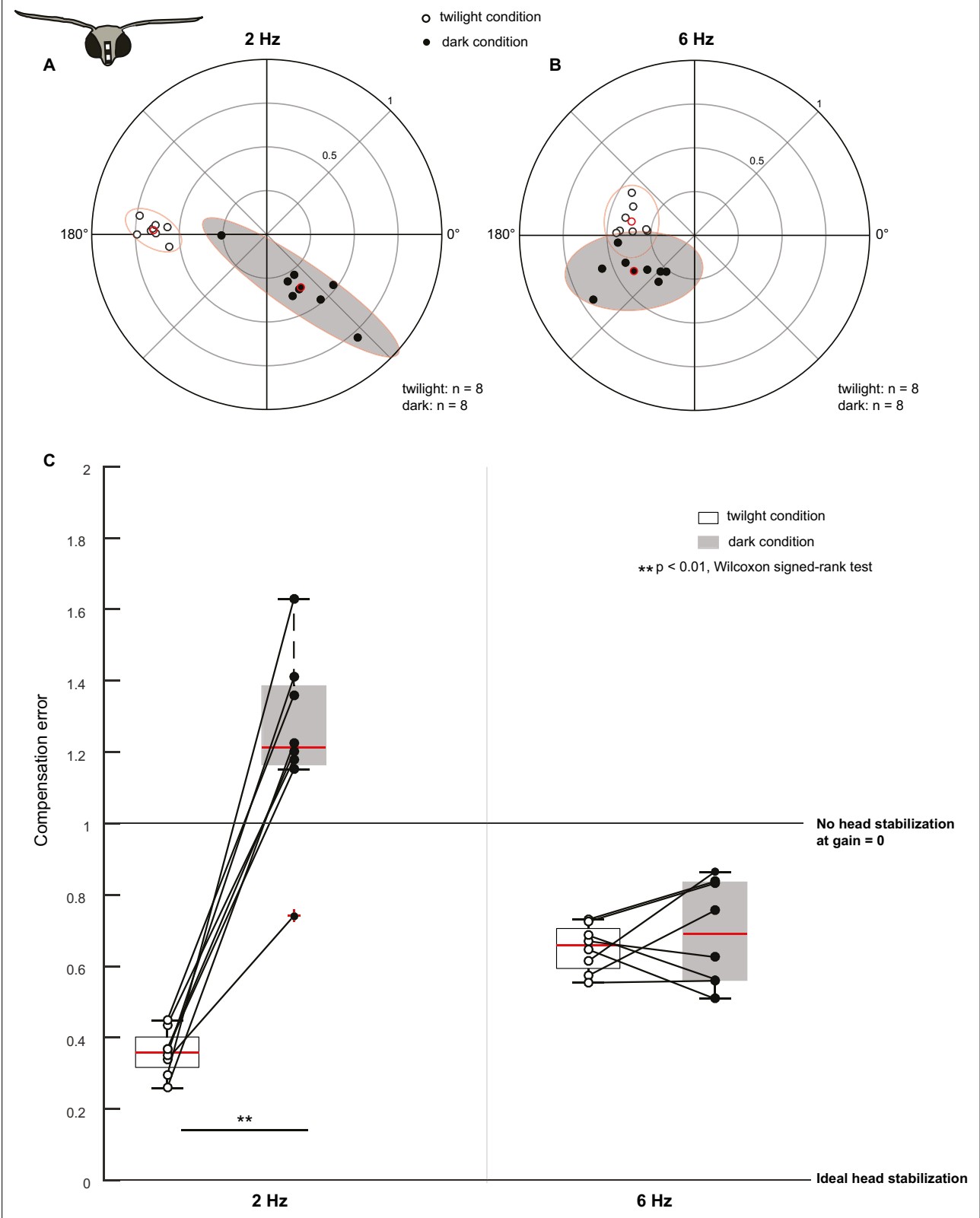

**Figure 2.** Visual feedback mediates head stabilization. (**A, B**) Polar scatter plots (with 95% confidence ellipses) of head roll response to body roll frequency of 2 Hz (**A**) and 6 Hz (**B**) in control moths. In each plot, the datapoints of the twilight condition are marked by open circles, the datapoints of the dark condition are marked by filled circles and their mean is highlighted in red. (**C**) Boxplots comparing compensation error between twilight and

*Figure 2 continued on next page*

*Figure 2 continued*

dark (filled boxplots) trials at 2 Hz (left two boxplots) and 6 Hz (right two boxplots). The datapoints are scattered on the plots. Datapoints representing the same animal in twilight and dark trials are connected with lines.

The online version of this article includes the following figure supplement(s) for figure 2:

**Figure supplement 1.** *Figure 2* is repeated with an alternative way of phase computation.

**Figure supplement 2.** *Figure 2A* is replotted with color gradient of compensation error.

**Figure supplement 3.** Scatterplots of gain at different stimulus amplitudes.

the twilight condition, the gain values remained largely unaltered with a slight phase shift in the *flagella-clipped* moths in comparison to the *flagella-intact* ones from a circular median of 177.6° to 174.0° (open circles, *Figures 2A and 3A*, *Figure 3—figure supplement 3* – Cases 1 and 3). Under the dark condition, the gain was very low in *flagella-clipped* moths with a median of 0.1 (filled circles, *Figure 3A*, *Figure 3—figure supplement 1E,G*, *Figure 3—figure supplement 3* – Case 4). Because the gain values are very low, the phase values were uninterpretable in this context. Unlike in the *flagella-intact* group at 2 Hz under dark conditions, the variability of the compensation error was very low under *flagella-clipped* moths (*Figure 3E*, filled circles, *Figures 2A and 3A*, *Supplementary file 1a*). The response (open and filled circles, *Figure 2A*) of *flagella-intact* moths at 2 Hz in the dark could, therefore, potentially result from antennal mechanosensory input, in turn influencing compensatory head movements at the low frequency in the dark.

At 6 Hz (*Figure 3E*), the compensation error was significantly different between the *control* (twilight: 0.7±0.02, dark: 0.7±0.05) and *flagella-clipped* (twilight: 0.8±0.03, dark: 1.0±0.03) groups in both twilight (p=0.0064) and dark conditions (p=0.0011) (*Supplementary file 1c*, 5, 6). Under twilight conditions, we observed a shift in phase in the *flagella-clipped* moths compared to the *flagella-intact* group from a circular median of 173.7° to 127.8° although the gains were similar (open circles, *Figures 2B and 3B*, *Figure 3—figure supplement 3* – Cases 1 and 3). Comparable to the responses observed at 2 Hz, the gain (median: 0.1, filled circles, *Figure 3B*, *Figure 3—figure supplement 1H*) and variability (*Figure 3E*, *Supplementary file 1a*) were lowest in *flagella-clipped* moths under dark conditions. These observations suggest that at 6 Hz, antennal mechanosensory input may be critical for head stabilization. Despite reduced antennal mechanosensory feedback, there are some indications of head stabilization under twilight conditions (compare open circles to filled circles, *Figure 3B*). This again suggests that visual feedback plays a role at this frequency. However, the compensation error of *flagella-clipped* moths under twilight condition at 6 Hz is significantly greater in comparison to 2 Hz (p=0.0156, Wilcoxon signed-rank test) because there is a shift in phase from a circular median of 174° to 127.8°. However, the gain values are similar (*Figure 3—figure supplement 3*, open circle: *Figure 3A, B*). Thus, the response due to visual feedback is reduced during faster body rolls.

Does restoring the mechanical load of flagella rescue head stabilization in the flagella-clipped moths? To determine this, we repeated the experiment in moths with reattached flagella. After reattachment of flagella, the head stabilization was similar to the performance of the moths with intact antennae, for all conditions. For both 2 and 6 Hz rotations, the compensation error in *flagella-reattached* moths (*Figure 3C–E*, *Figure 3—figure supplement 1I-L*) was not significantly different from the *flagella-intact* group under twilight and dark conditions (see *Supplementary file 1c*, 3–6 for statistical comparisons). Similar to the response of the *flagella-intact* group at 2 Hz, we observed a phase-shift between the twilight and dark conditions in the *flagella-reattached* moths from a circular median of 177° to 297.6° at 2 Hz and from 149.4° to 216° at 6 Hz (*Figure 3—figure supplement 3* – Cases 5 and 6, *Figures 2A and 3C*). For 6 Hz rotations in the dark condition (*Figure 3E*), the compensation error of the *flagella-reattached* moths differed significantly from the *flagella-clipped* group (*Supplementary file 1c*. 6). The

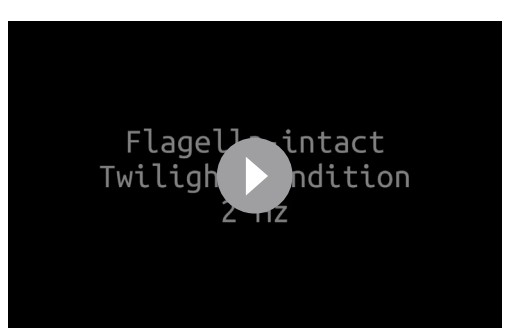

**Video 1.** Role of visual feedback in head stabilization. Representative videos of *flagella-intact* moths under twilight and dark conditions at 2 Hz.

https://elifesciences.org/articles/78410/figures#video1

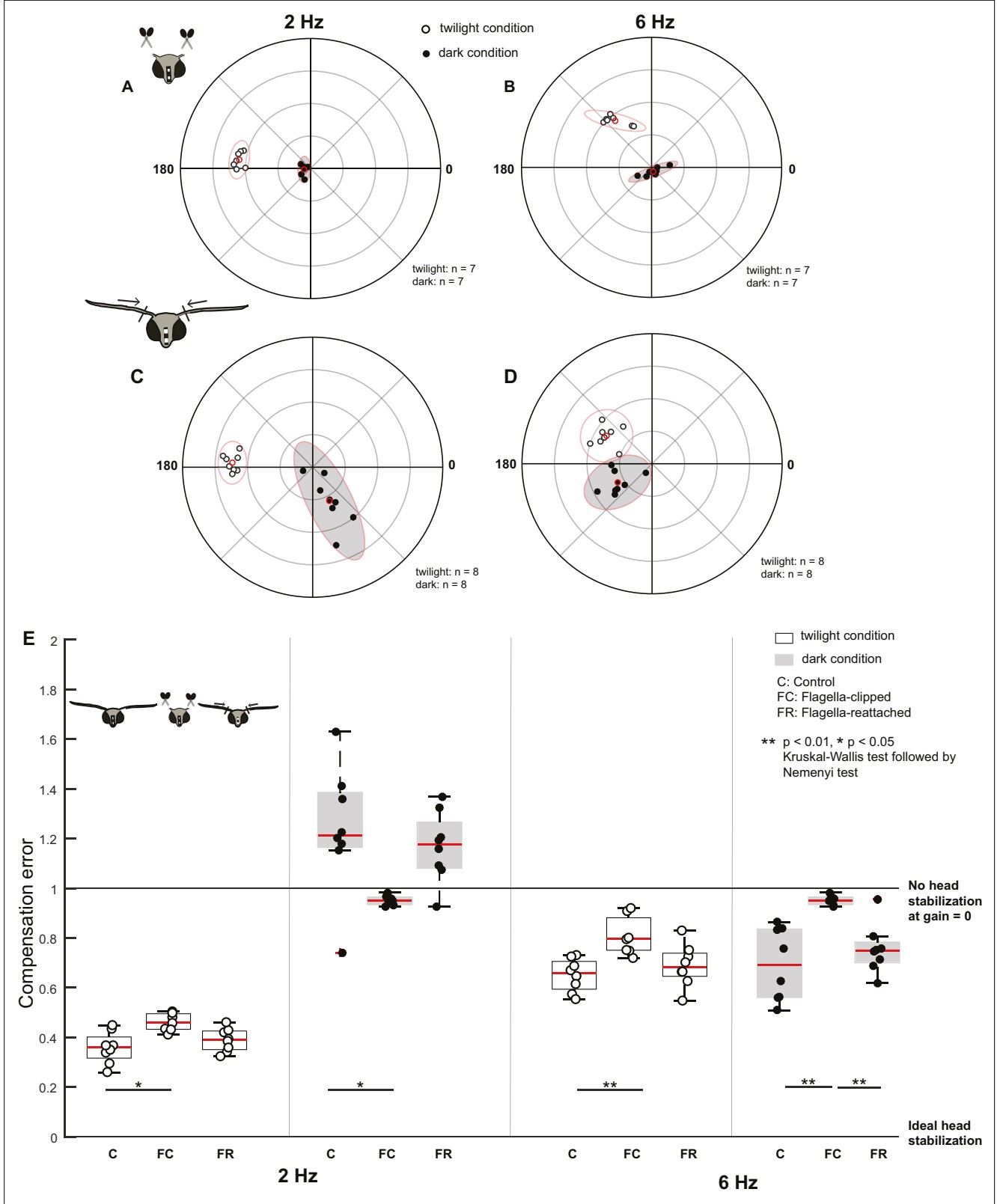

**Figure 3.** Antennal feedback is required for head stabilization. (**A–D**) Polar scatter plots (with 95% confidence ellipses) of 2 Hz (**A, C**) and 6 Hz (**B, D**) roll frequencies in *flagella-clipped* (**A, B**) and *flagella-reattached* moths (**C, D**). In each plot, the datapoints of the twilight condition are marked by open circles, the datapoints of the dark condition is marked by filled circles, and the mean is highlighted in red. (**E**) Boxplots comparing compensation error

*Figure 3 continued on next page*

*Figure 3 continued*

between *control* (C), *flagella-clipped* (FC) and *flagella-reattached* (FR) moths for twilight and dark (filled boxplots) trials at 2 Hz (left six boxplots) and 6 Hz (right six boxplots). Note that the control dataset in this plot is used from the previous experiment (*Figure 2C*).

The online version of this article includes the following figure supplement(s) for figure 3:

**Figure supplement 1.** Representative time-domain plots in twilight and dark conditions (gray shaded) at 2 Hz (left column) and 6 Hz (right column) in *flagella-intact* (**A–D**), *flagella-clipped* (**E–H**), and *flagella-reattached* (**I–L**) groups.

**Figure supplement 2.** Coherence estimates of different treatments.

**Figure supplement 3.** List of median values of gain and phase for the different conditions of *flagella-intact, flagella-clipped,* and *flagella-reattached* moths.

coherence estimates were also highly variable in *flagella-clipped* moths under the dark conditions in comparison to the corresponding *flagella-intact* and *flagella-reattached* groups (*Figure 3—figure supplement 2A*, *Supplementary file 1b*). Together, these experiments suggest that antennal mechanosensory input is required for head stabilization in hawkmoths primarily at higher frequency, besides some contribution at lower frequency as well (*Video 2*). Our results also indicate some contribution of visual feedback at higher frequency rotation.

## The role of Johnston's organs in head stabilization

Johnston's organs respond to extremely subtle deformations and vibrations of the flagellum relative to pedicel (*Lin and Leonardo, 2017*). Their role in flight stabilization has been highlighted in previous studies (*Olberg et al., 2007*; *Paulk and Gilbert, 2006*), and hence we specifically tested their contribution to head stabilization. In the experiments described above, the ability of the *flagella-clipped* moths to stabilize their heads was reduced although visual feedback may offset this effect to a certain extent at 2 Hz and to a lesser extent at 6 Hz. We therefore tested the hypothesis that this effect in moths with removed flagellar load was specifically due to the reduction of mechanosensory feedback from the Johnston's organs. To reduce feedback from the Johnston's organ, we glued the pedicel-flagellar joint of moths in which the Johnston's organ is embedded. Data from these experiments were compared with *sham-treated* moths in which the ~2nd–3rd annuli in the flagella were glued, rather than the pedicel-flagellar joint (*Figure 4A–E*).

Under twilight and dark conditions for 2 Hz roll stimulus, the compensation error between the *sham* (twilight: 0.4±0.02, dark: 1.2±0.08) and *Johnston's organ-glued (JO-glued)* groups (twilight: 0.4±0.01, dark: 1±0.04) was not significantly different (*Figure 4E*, p=0.5358, p=0.0831, *Supplementary file 1c*, 7, 8). However, similar to the *flagella-clipped* group, the *JO-glued* moths exhibited very low gain (median: 0.11) under dark conditions and therefore, their phase values in this condition are not interpretable (filled circles, *Figures 3A and 4C*). Also, similar to the *flagella-intact* group, the *sham* group showed high variability in the dark conditions, and phase values shifted from 178.5° in twilight condition to 306.3° in dark condition (*Supplementary file 1a*, *Figure 4E*, filled circles, *Figure 4A*). At 6 Hz in both twilight and dark conditions, the compensation error differed significantly between the *sham* (twilight: 0.6±0.03, dark: 0.6±0.06) and *JO-glued* (twilight: 0.9±0.05, dark: 1±0.04) groups (*Figure 4E*, p=0.0097, p=0.0055, *Supplementary file 1c*, 9, 10). Thus, we observed similar effects in *JO-glued* and *flagella-clipped* moths (*Figures 3A–B, E and 4C–E*), suggesting that feedback from the Johnston's organ affects head stabilization. The coherence values were variable in the *JO-glued* group in comparison to those obtained in the *sham* group under dark conditions (*Supplementary file 1b*, *Figure 3—figure supplement 2*).

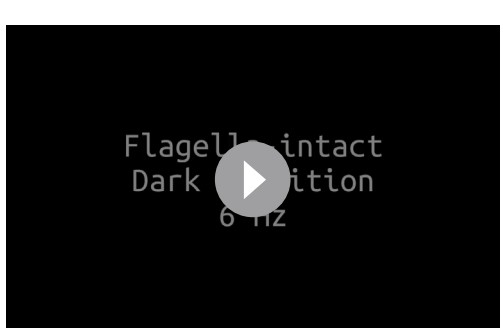

**Video 2.** Role of antennal feedback in head stabilization. Representative videos of *flagella-intact* and *flagella-clipped* moths under dark conditions at 6 Hz.

https://elifesciences.org/articles/78410/figures#video2

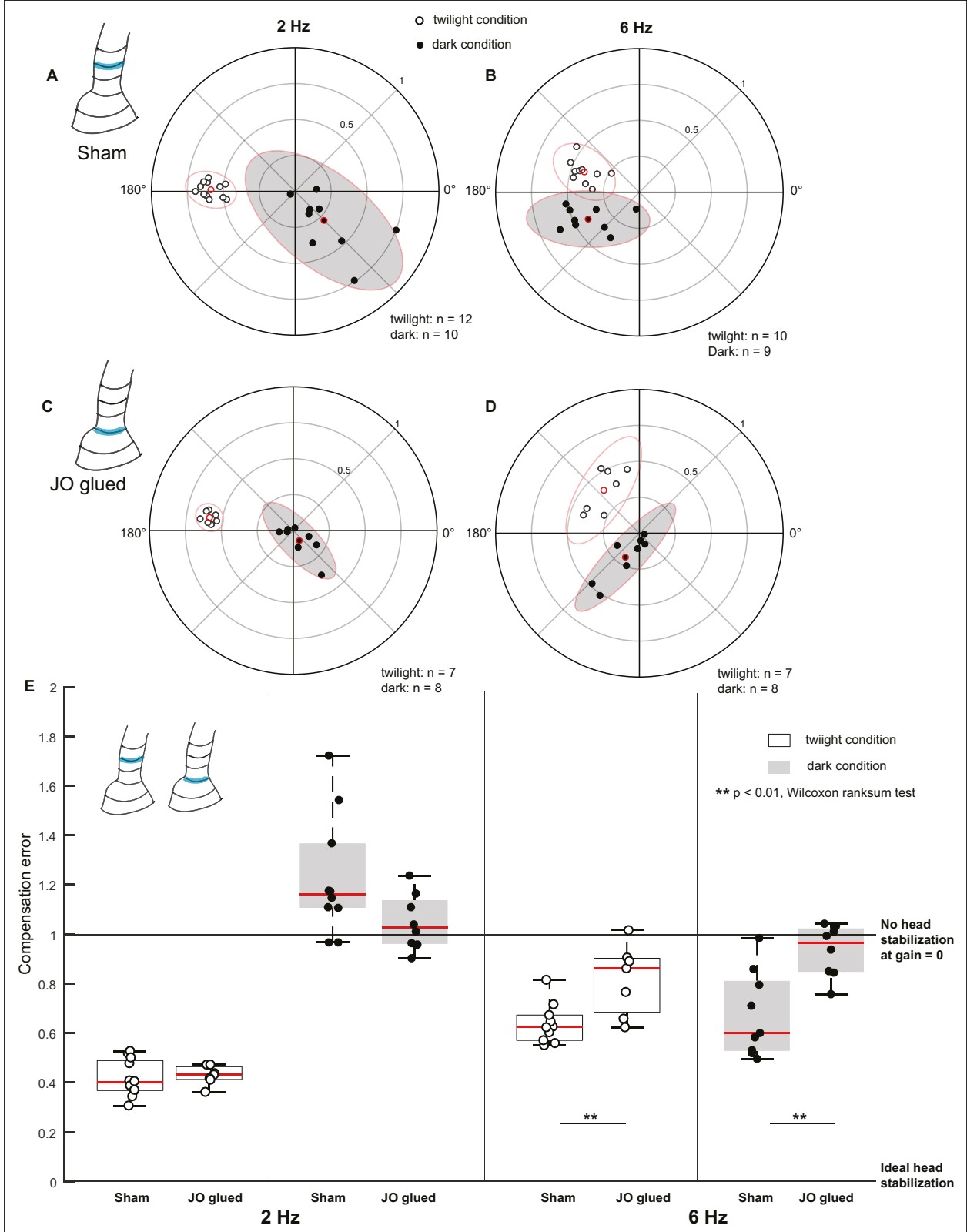

**Figure 4.** Johnston's organ contributes to head stabilization. (**A–D**) Polar scatter plots (with 95% confidence ellipses) of 2 Hz (**A, C**) and 6 Hz (**B, D**) roll frequencies in *sham* (**A, B**), and *Johnston's organ-glued (JO-glued)* (**C, D**) moths. In each plot, the datapoints of the twilight condition is marked by open circles, the datapoints of the dark condition is marked by filled circles and the mean is highlighted in red. (**E**) Boxplots comparing compensation error between *sham* and *JO-glued* moths for twilight and dark trials at 2 Hz (left four boxplots) and 6 Hz (right four boxplots).

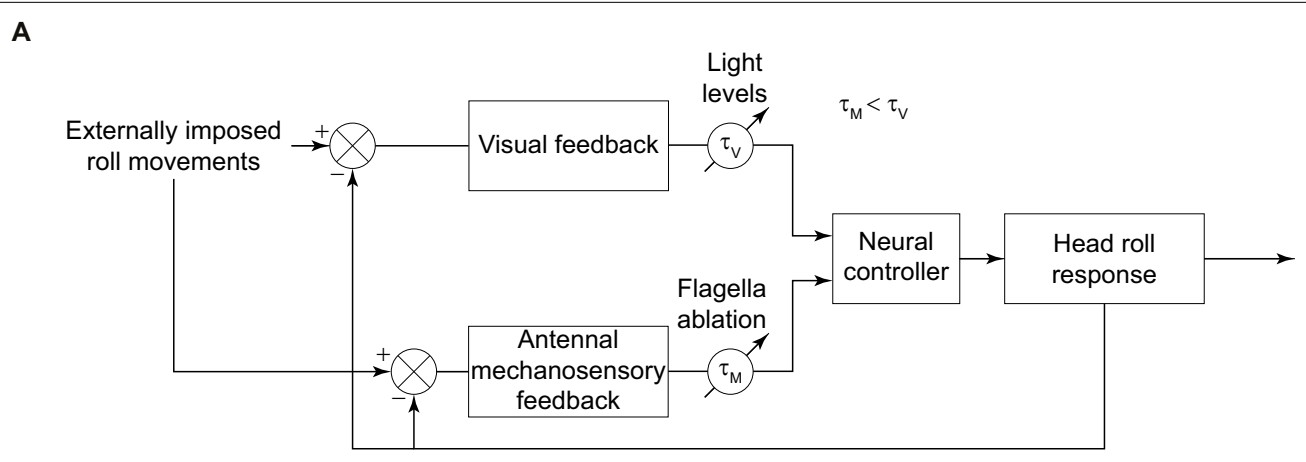

**Figure 5.** Summary figure. A block diagram illustrating the role of visual and antennal mechanosensory feedback in the head stabilization response. The visual and antennal mechanosensory feedback sections of the loop have time delays ($\tau_v$, $\tau_m$) which vary based on illumination levels and flagella ablation state, respectively. These responses combine to elicit the head roll response.

## Combinatorial effects of visual and antennal mechanosensory feedback

In the experiments presented here, maximal deviation in phase occurs when either visual feedback is reduced under dark conditions at low-frequency roll stimuli (compare Case 2 with the control Case 1; *Figure 3—figure supplement 3*) or else when antennal mechanosensory feedback is reduced by clipping flagella or gluing JO at high-frequency roll stimuli (compare Case 3 with Case 1; *Figure 3— figure supplement 3*). When antennal feedback was restored by reattaching the flagellum, both gain and phase values were similar to the intact antenna case (e.g. compare Case 1 with Case 5, and Case 2 with Case 6; *Figure 3—figure supplement 3*). These data show that visual feedback is essential at low-frequency roll whereas antennal mechanosensory feedback is crucial at high-frequency roll maneuvers. If both visual and antennal mechanosensory feedback are reduced or eliminated, gain is reduced to zero or near zero values indicating that head roll response is absent (*Figure 3—figure supplement 1G*, H), under which conditions, the phase values are uninterpretable. On the other hand, when *flagella-intact, flagella-reattached*, and *sham* moths (*Figures 2–4*) underwent low-frequency roll in dark conditions, the compensation error exceeded 1 and the gain was non-zero, indicating that the head response was elicited but in the wrong direction due to the phase-shift from the ideal stabilization scenario (filled circles in *Figures 2A–4A*). In the corresponding *flagella-clipped* and *JO-glued* moths, the gain values were very low indicating that no response was elicited (filled circles; *Figures 3A and 4C*) in absence of antennal mechanosensory feedback, resulting in compensation error values near 1 (*Figures 3E and 4E*).

## Restricted head movements cause loss of flight performance in freely flying moths

Together, the above experiments on tethered moths indicated that mechanosensory feedback from the antennal Johnston's organ is involved in head stabilization, in addition to the visual feedback from its retina (*Figure 5*). We next hypothesized that flight impairment observed in previous studies in the flagella-clipped hawkmoths, *Manduca sexta* (*Sane et al., 2007*), may arise from inability of these hawkmoths to stabilize their head during flight. If that is the case, then we expect moths with restricted heads to show similar flight impairments as those observed in the *flagella-clipped* moths. We filmed the free-flight trajectories (see Materials and methods) of moths whose head movements were restricted. We compared these with *control* moths in which the head was free to move, and with *sham* moths which underwent the same procedure as experimental moths but without head restriction. In our experiments, out of 12 animals in each group, 11 *control* animals, 12 *sham*-treated, and 8 experimental *head-restricted* moths produced free-flight behavior (comparison with internal control: *Figure 6—figure supplement 1A*, B). The flight duration of *control* (73±17.0 s) and *sham* (34±9.3 s) groups were significantly longer compared to the moths with restricted head-motion

(1.5±0.7 s) (*Figure 6A*, p=0.0001, *Supplementary file 1c*, 11). We also observed a difference in the flight behavior of the *head-restricted* moths in comparison to the *control* and *sham* moths. Whereas the *control* and *sham* moths took off smoothly and pitched up to a considerable height, the *head-restricted* moths were unable to pitch up, and consequently stayed close to the ground usually touching it, and became unstable as soon as they became airborne (*Video 3*). The frequency of these moths (1±0.1 per second) colliding with the walls of flight chamber was significantly greater than in *control* (0.4±0.05 per second) and *sham* (0.3±0.05 per second) moths, thus indicating impairment in their flight performance (*Figure 6B*, p=0.0002, *Supplementary file 1c*, 12).

In the free-flight assay described above, the moths were not presented with any specific stimuli to generate flight activity. Thus, the flight duration depended entirely on when the moth voluntarily chose to stop flapping and land on a surface, which caused substantial variability in the free-flight data on control moths. In the head-restricted moths however, flight duration is significantly lower compared to all other cases, implying that *head-restricted* moths cannot fly in a controlled manner and collide more frequently per unit time, when compared to moths in the *control* and *sham* group (*Figure 6—figure supplement 1*).

## Discussion

Compensatory head movements are crucial for stability in free-flight (*Figure 6A–B*). The data presented in this paper show that such compensatory head roll in tethered flight is mediated via a combination of visual and antennal mechanosensory feedback in the hawkmoth, *D. nerii* (*Figure 5*).

### Role of visual feedback in flight and gaze control in hawkmoths

In diurnal insects, visual feedback plays a key role in eliciting compensatory head movements during complex maneuvers (*Hengstenberg, 1993*; *Viollet and Zeil, 2013*). However, at lower light levels, the visual system transduces at slower rates thereby imposing severe constraints on insects, especially during aerial maneuvers (*Warrant and Dacke, 2011*). Previous studies on freely flying hawkmoths have documented the role of visual feedback in flower tracking by quantifying their responses at different light intensities (diurnal hawkmoths, *Macroglossum stellatarum Dahake et al., 2018*; *Stöckl et al., 2017*), crepuscular/nocturnal hawkmoths, *M. sexta* (*Sponberg et al., 2015*; *Stöckl et al., 2017*), and *Deilephila elpenor* (*Stöckl et al., 2017*). Electrophysiological studies from the wide-field motion-sensitive neurons of nocturnal hawkmoths suggest spatial and temporal tuning of their neurons to their ecological niche (*Theobald et al., 2010*). Together, these studies show that despite the handicap of slow visual transduction rates, hawkmoths can fly or perform aerial maneuvers under low light levels.

We conducted this study on the Oleander hawkmoth *D. nerii* primarily because it typically flies in low light and we used similar light levels as used in the flower tracking studies of crepuscular and nocturnal hawkmoths *Manduca* and *Dielephila* (300 and 0.3 lux) (*Sponberg et al., 2015*; *Stöckl et al., 2017*). Our data show that gaze stabilization in *D. nerii* requires visual feedback at low temporal frequency, and antennal mechanosensory feedback at higher temporal frequencies at which purely visual means of gaze stabilization may be constrained due to the requirement for both spatial and temporal pooling of photons to form images or derive motion information. Specifically, we hypothesize that gaze stabilization is enhanced by a fast response via the antennal mechanosensory feedback loop which ensures that the head remains sufficiently stable to enable temporal and spatial integration of photons by the superposition eyes.

### Role of antennal mechanosensory feedback in control of head stabilization and flight

We used two treatments to reduce the antennal mechanosensory feedback from the Johnston's organs. In one case, we clipped off flagella to cause mechanical unloading of Johnston's organ and in the second case, we glued the pedicel-flagellar joint. In both cases, we observed similar impairment of head stabilization (*Figures 3E and 4C*). Moreover, regluing the flagella and thus restoring Johnston's organ feedback caused recovery of head stabilization. These results suggest that Johnston's organ is the key antennal mechanosensory organ involved in head stabilization. In these experiments, *flagella-clipped* moths were more strongly affected than the *JO-glued* moths, indicating some

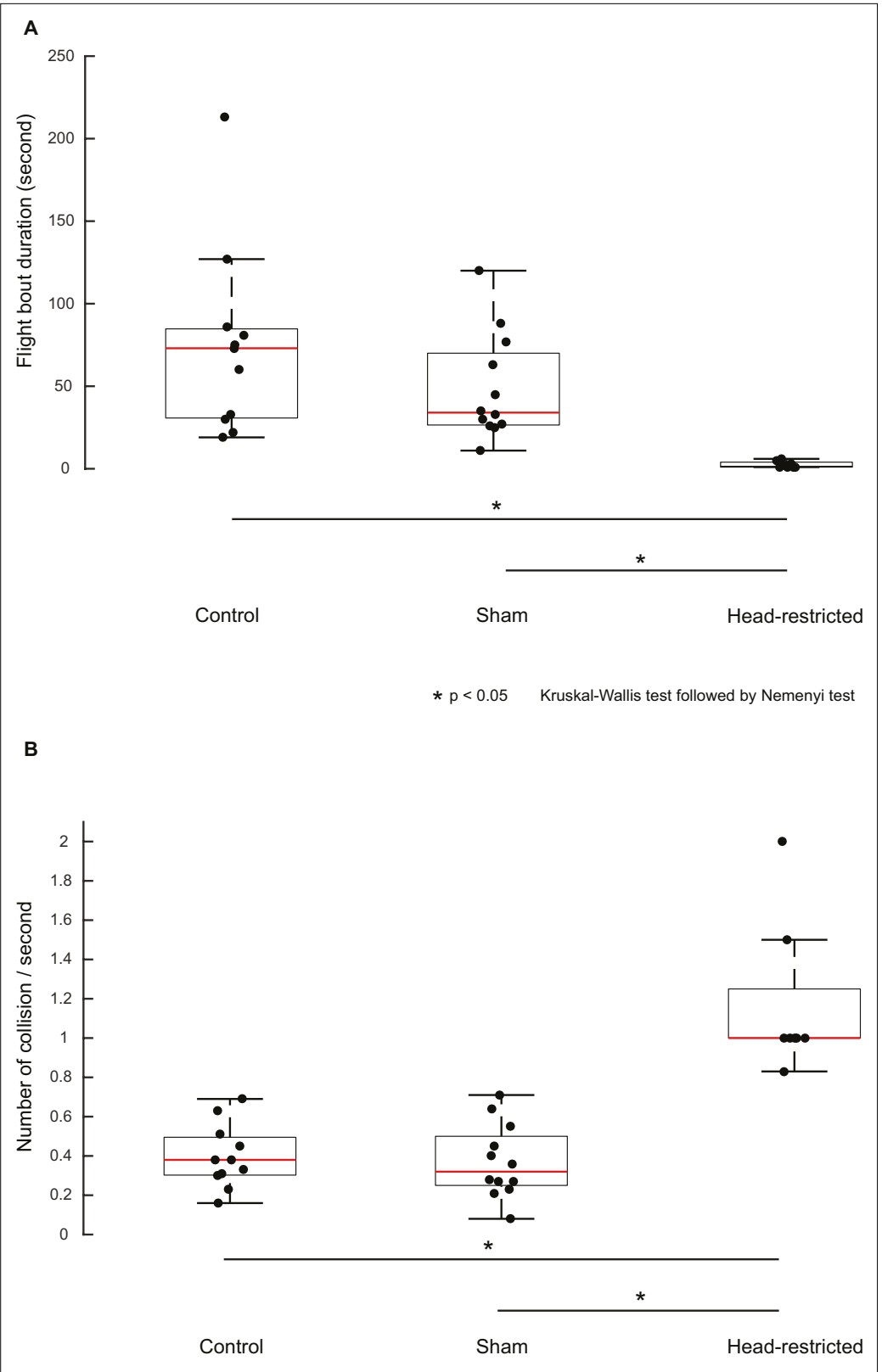

**Figure 6.** Restricting head movements impairs flight stability. Boxplots comparing flight bout duration (**A**) between take-off to landing and collision frequency (**B**) on side walls of the flight arena between *control*, *sham*, and *head-restricted* moths in free-flight.

*Figure 6 continued on next page*

*Figure 6 continued*

The online version of this article includes the following figure supplement(s) for figure 6:

**Figure supplement 1.** Assessment of free-flight parameters with their internal controls.

**Figure supplement 2.** Free-flight set-up and efficacy test of head-restriction treatment.

residual feedback from Johnston's organ after gluing the pedicel-flagellar joint but not enough to ensure proper head stabilization (**Figure 4**).

Although previous experiments have indicated that antennal mechanosensory feedback is crucial for maintaining stable flight (**Dahake et al., 2018**; **Sane et al., 2007**), the precise link between antennal mechanosensors and flight has remained largely unexplored. Antennae sense airflow in diverse insects (**Gewecke, 1974**), and moths and bees modulate antennal positioning during flight (**Roy Khurana and Sane, 2016**; **Natesan et al., 2019**). Antennae help in maintaining headwind orientation (**Fuller et al., 2014**), mediating abdominal flexion (**Hinterwirth and Daniel, 2010**), and flight balance (**Dahake et al., 2018**; **Sane et al., 2007**). In this study, we show that Johnston's organ feedback affects compensatory head roll. Moreover, restriction of head movement causes impediment in flight performance. Thus, flight impairment due to loss or reduction of Johnston's organ input may result in part from their inability to control compensatory head movements.

## Head stabilization in tethered hawkmoths is undercompensated

In our experiments, the gains for compensatory head movements ranged from 0.2 to 0.6 depending on the frequency of the forced body rotations under twilight condition, which meant that the head was not fully stabilized in its horizontal orientation. We ruled out the hypothesis that such low gain results from the stimulus amplitude exceeding anatomical constraints of the moth's neck motor system limiting the angular range of head roll rotations, as the gain was less than 1 also for low-amplitude stimuli (**Figure 2—figure supplement 3**). Previous studies in diverse insects including flies (**Hengstenberg et al., 1986**), locusts (**Hensler and Robert, 1990**), and wasps (**Viollet and Zeil, 2013**) show that head stabilization does not fully compensate for the imposed stimulus. Undercompensated head stabilization for low stimulus amplitudes has also been previously observed in blowflies (**Hengstenberg et al., 1986**), suggesting that compensatory head rotations are only one of several components contributing to optomotor reflexes and are not the only mechanism involved in maintaining a level gaze. In free-flight, perhaps, both compensatory head rotations and body rotations would contribute to a level gaze. It remains to be seen if such undercompensation exists in freely flying conditions and has a clear function in flying insects. Alternatively, gaze stabilization in insects may be inherently undercompensated, as insects have been reported to not respond to a stable pattern (detailed discussion in **Hengstenberg et al., 1986**).

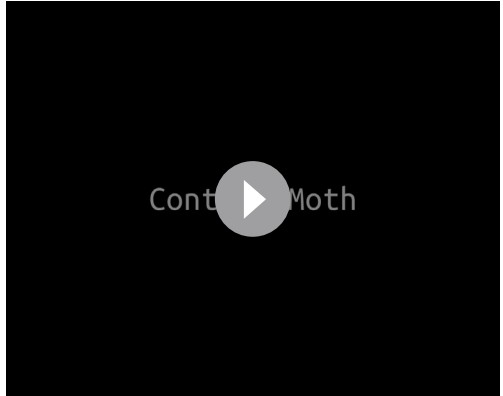

**Video 3.** Role of head movements in free-flight behavior of hawkmoths. Representative videos of *control*, *sham* moths showing stable flight, example of a side-wall collision and *head-restricted* moths showing unstable flight, crawling, and skidding locomotion.
https://elifesciences.org/articles/78410/figures#video3

## Multisensory control of reflexes

A key result of these experiments is that, just as flies integrate haltere and visual feedback for head stabilization (**Hardcastle and Krapp, 2016**; **Hengstenberg, 1993**; **Schwyn et al., 2011**), the head stabilization reflex in moths also requires the integration of visual and antennal mechanosensory inputs. In absence of a detailed frequency analysis, we were unable to determine if the combinatorial effect of visual and antennal mechanosensory is linear or non-linear. A consistent result across these studies is that visual feedback influences head stabilization behavior in a lower frequency regime, whereas mechanosensory feedback – whether from antenna or halteres – influences head stability at a higher frequency regime. In flies, halteres detect fast angular

velocities of thorax rotation and elicit head stabilization with a gain of 0.1 around 50°/s, which reaches a maximum of about 0.75 at around 1000°/s (*Hengstenberg, 1993*; *Sherman and Dickinson, 2003*). Similarly, in moths the antennal mechanosensory feedback affects tracking only at high frequencies (>2 Hz) of flower oscillations, whereas visual feedback is key at lower frequencies (*Dahake et al., 2018*). Indeed, the frequency range in which visual and antennal inputs influence head stabilization in the present study is similar to the frequency range reported in studies on freely flying moths (*Dahake et al., 2018*). In flies, some neck motor neurons require simultaneous activation of halteres and visual pathways for eliciting spikes, thus setting up a gating mechanism which would enable compensatory head movements in a dynamic range that is in between the maximum sensitivities of the motion vision and haltere system (*Huston and Krapp, 2009*; *Theobald et al., 2010*). Whereas head wind and visual stimuli together elicit spikes in the ventral cervical nerve motorneuron (VCNM) of the neck motor system, they do not individually fire action potentials in VCNM in the blowfly, *Calliphora vicinia*. In that study, the removal of the arista of the antenna abolished this VCNM response (*Haag et al., 2010*). Thus, flies may also possess a similar gating mechanism in the visual and antennal pathways that acts at the level of neck motor neurons. It is likely that similar gating exists in the visual and antennal mechanosensory pathways in non-Dipteran insects.

### Head wobble

In all the recordings of head stabilization, we observed small-amplitude head oscillations (or head wobble; *Chatterjee et al., 2022*) at approximately 12 Hz, which did not correspond to the wingbeat or any other frequency that we have investigated. These regular oscillations are a predicted outcome of the multisensory negative feedback loop that we have proposed, and depends on the lag in the sensory acquisition (*Chatterjee et al., 2022*), or its translation to motor commands in addition to the mechanical properties of head-neck apparatus. Alternatively, it may result from active head movements that the moth performs to ensure active visual feedback (*Cellini and Mongeau, 2020*; *Stamper et al., 2012*). Such head movements may help increase the dynamic range of the visual pattern movement speeds encoded by the insect motion-vision system (*Windsor and Taylor, 2017*). A formal model of the head stabilization system requires a full systems characterization of the mechanical properties of the head-neck system, which will be explored in a future study.

### Conclusion

Our experiments demonstrate that head stabilization in hawkmoths requires the combined feedback from the compound eyes and antennal mechanosensors. Under sufficient lighting and during slower turns, hawkmoths are able to stabilize their head using visual feedback. However, under dark conditions or during faster turns, the role of antennal mechanosensory feedback from the Johnston's organs is also essential. In absence of gaze stabilization, flight control in freely flying hawkmoths is severely impaired. Taken together, these results show that the loss of flight control due to reduction of antennal mechanosensory feedback may result at least partly from an inability to stabilize gaze. Thus, our results point to the importance of visual and mechanosensory integration in head stabilization behavior of insects. In Diptera, this has been shown to involve mechanosensory feedback from halteres in addition to the neck proprioceptive feedback from prosternal organs and other potential contributions from mechanosensory feedback in legs, wings, and abdomen. The data presented here show that for non-Dipteran insects such as hawkmoths, mechanosensory feedback is derived partly from the antennal Johnston's organs.

## Materials and methods
### Resource availability
#### Lead contact
Further information and requests for resources and reagents should be directed to and will be fulfilled by the lead contact, Prof. Sanjay P Sane (sane@ncbs.res.in).

#### Materials availability
This study did not generate new unique reagents.

## Data and code availability

The datasheets and codes are deposited in Mendeley data and available in the following link: https://data.mendeley.com/datasets/2trxj9gwsw/draft?a=89356c26-e581-40c6-935d-c1d4a0401074.

Any additional information required to reanalyze the data reported in this paper is available from the lead contact upon request.

## Experimental model and subject details

### Moth breeding

We used adult ~1-day post-eclosion *D. nerii* bred in a greenhouse for experiments. The temperature of the greenhouse was maintained at ~28°C. The adult male and female moths were released in a netted chamber in the greenhouse for mating and egg-laying. Larvae of all instars were reared on a natural diet of *Nerium oleander* leaves inside mesh-topped boxes in the greenhouse. The pupae were initially kept in sawdust inside the mesh-topped boxes after which they were transferred to a netted cage to allow emergence. Adult *D. nerii* begin flying around sunset and remain active through the night (operating light levels: ~0.05–250 lux, unpublished observations).

## Method details

### Treatment procedure

We anesthetized moths by keeping them in –20°C for about 8 min. The thorax of each moth was descaled before attaching to it a 3 mm neodymium magnet using cyanoacrylate glue (Super glue, EVO BOND, Evo bond group, Taiwan). There were five experimental groups: *control*, *flagella-clipped*, *flagella-reattached*, *JO-glued,* and its corresponding *sham* procedure.

The *control* group was anesthetized and tethered just as the experimental groups, but underwent no further treatment. In the *flagella-clipped* moths, we cut both the left and right flagella around the 3rd–4th annuli thereby reducing the mechanical load on the Johnston's organs. In the *flagella-reattached* moths, we cut and then reattached the excised flagella to their stumps using cyanoacrylate glue. Finally, for the *JO-glued* group and its corresponding *sham* treatment, we kept the moths on a metal-plate placed atop an ice-bath throughout the treatment procedure (~30 min). In both groups, the head and the base of the antennae were descaled to expose the pedicel-flagellar joint. Cyanoacrylate glue was applied to the pedicel-flagellar joint in the *JO-glued* group. In the *sham* group, glue was applied approximately at the junction between 2nd and 3rd annulus of the flagellum. This ensured that the gluing procedure itself did not affect head stabilization. All the groups were given about half an hour of recovery post-treatment before starting the experimental assay.

### Behavioral protocol

We mounted the moths on a tether which was attached to the shaft of a stepper motor (NEMA 23, 1.8° step angle, 10.1 kg·cm torque). The tether was bent into a U-shape to ensure that the longitudinal axis of the moth aligned with the shaft axis (*Figure 1A*). To achieve smooth motion, we used a low-noise microstepping driver (Leadshine ND556) with a 125 microstep resolution (i.e. 25,000 steps/revolution) in all our experiments. Using a microcontroller board (PJRC Teensy 3.5, 120 MHz), we generated stimuli of desired frequencies. We used split magnets for tethering the moths, one on the dorsal thorax and the other at the end of the tether rod. In this procedure, neodymium magnets were split using a flat file and the two half-magnets were joined to prevent slipping of the magnets about the yaw axis. The tether was rotated with a sinusoidal stimulus of peak-to-peak amplitude of 60° (±30°) and the moth head was aligned with the shaft of the motor to ensure that the roll stimulus was around the longitudinal axis going through the head, approximately aligned with the pivot point of the neck motor system. We ensured that the moths were actively flapping their wings during the experiments.

We placed a high-speed camera (V.611, Phantom) frontally and filmed the moths at 1200 frames/s using 200 μs exposure. We painted the region between the two eyes along the midline of the moth head with black acrylic color (*Figure 1B*, icon in *Figure 2A*), put small white papers on it and the tether rod for easy digitization. IR LEDs (wavelength: 850 nm, Mouser: 720-SFH4715) were mounted around the lens for additional lighting. All the groups were filmed in the laboratory environment (*Figure 1—figure supplement 1*) in two ambient light conditions under which the moths are typically active (*twilight*: ~250 lux; *dark*: <0.01 lux). We used a lux meter (center 337; range: 0.01–40,000 lux) to

measure light intensity around the moth. These experiments were carried out in the ambient setting of the laboratory which contained both 3D and depth cues (for details, see *Figure 1—figure supplement 1*). The surrounding visual features were kept constant across all the experiments. Each experimental treatment was always compared with its corresponding control (i.e. intact or sham treatment under same visual background). Because our conclusions are based only on the difference between the control and experimental scenario regardless of the actual visual cues presented, our results are expected to be robust for any arbitrary visual environment.

We rotated the tether at two frequencies: 2 Hz (low) and 6 Hz (high) for 8 cycles each, corresponding to mean angular speeds of 240°/s (low) and 720°/s (high), respectively. The maximum stimulus frequency our stepper motor could reliably generate was 6 Hz. Frequencies of 2 Hz and 6 Hz are in alignment with the dynamic range of stimuli provided in previous studies of flower tracking in other hawkmoth species. In the diurnal hawkmoth, *M. stellatarum* (*Dahake et al., 2018*), antennal mechanosensory feedback was reported to play a significant role in flower tracking in the frequency range of 2–5 Hz. Further, under similar light levels used in our study, flower tracking was affected between the light levels in another study in the crepuscular hawkmoth, *M. sexta*, approximately in the frequency range of 2–8 Hz (*Sponberg et al., 2015*). At the beginning and end of each stimulus period, a quarter of the sinusoidal oscillation of the tether movement was replaced with a linear ramp to avoid excessive accelerations and thus, potential shifts of the magnetic tether. The corresponding time periods of the responses were excluded from our analysis. The sequence of the light conditions and the roll frequencies tested were randomized across all moths.

## Quantification and statistical analysis
### Analysis
The videos were digitized using a custom C++ program in OpenCV library. We then calculated the angles between the thorax and frame vertical ($\theta_{thorax}$) and between head and frame vertical ($\theta_{head}$) in MATLAB. From the above angles, we computed the angle between the thorax and head ($\theta_{head\text{-}thorax}$) (*Figure 1B*). The raw data were filtered using a 7th order low-pass Butterworth filter with cut-off frequency of 25 Hz, to eliminate the wing-beat-induced noise in the head movements which occurs at the wing-beat frequency of *D. nerii* (~30 Hz). The average amplitude at 2 Hz and 6 Hz was calculated by Fourier transform from the corresponding time series. We calculated the gain using the formula:

$$\text{Gain} = \frac{\text{Average amplitude of } \theta_{head-thorax}}{\text{Average amplitude of } \theta_{thorax}} \tag{1}$$

The phase difference (hereafter referred to as phase) between $\theta_{thorax}$ and $\theta_{head\text{-}thorax}$ was computed by cross-correlating the two time series in MATLAB. The time difference corresponding to the maximum correlation between the two time series was used for estimating phase. Phase (in radians) was computed from time difference using the equation:

$$\text{Phase} = 2\pi * \left(\text{frequency of oscillation}\right) * \left(\text{time difference}\right) \tag{2}$$

We also estimated the phase difference in *flagella-intact* moths based on a fast Fourier transformation (FFT) of the responses in MATLAB (*Zhivomirov, 2021*, *Figure 2—figure supplement 1*), where the complex output of FFT contains both magnitude and phase at each frequency. If the animal stabilizes its head perfectly (i.e. fully compensated head stabilization), the gain is 1 and the phase is 180° (*Figure 1C*). When the animal does not stabilize its head, the gain is 0 (*Figure 1C*). If the gain is <1, then head stabilization is undercompensated (*Figure 1D*). We represent gain and phase using polar scatter plots in which gain is represented in the radial axis and phase is depicted by the direction of each dot in the circular axis (*Figure 1E*). A cluster of points, representing the results obtained during a particular treatment, is enclosed by a 95% confidence ellipse (*Wang et al., 2015*; *Figure 1F*). Head-compensation error parameter ε was computed as the distance between the perfect head stabilization point and the experimental point from a trial (*Figure 1E*; for details, see supplementary information, also see *Figure 2—figure supplement 2*). In a polar scatter plot, the distance between two points $p_1$ and $p_2$ with polar coordinates $(r_1, \theta_1)$ and $(r_2, \theta_2)$ is given by

$$d = \sqrt{r_1^2 + r_2^2 - 2r_1 r_2 \cos\left(\theta_1 - \theta_2\right)}$$

Compensation error ($\varepsilon$) is defined as the distance between a point where there is perfect head stabilization and the point as obtained following an experimental manipulation. Thus, if $p_1$ depicts the point corresponding to perfect head stabilization with coordinates (1, 180°), the compensation error ($\varepsilon$) for the experimental point $p$ with coordinates $(r, \theta)$ is

$$\varepsilon = \sqrt{1 + r^2 - 2r\cos\left(180° - \theta\right)}$$

In this representation, the compensation error ($\varepsilon$) can range from 0 to 2 (for gain ≤1) with an $\varepsilon$ of 0 representing perfect head stabilization and corresponds to (r=1, θ=180°) in the polar-scatter plots (*Figure 1E*). The point corresponding to (r=1, θ=0°) assumes an $\varepsilon$ of 2 and as the error increases from 0 to 2, there is a reduction in the compensatory head roll. In the point corresponding to (r=0, θ=0°) which assumes an $\varepsilon$ of 1, there is no head movement relative to the thorax and therefore, no compensation.

The boxplots were generated in MATLAB using the following conventions:

> Central line – median
> Bottom edge of the boxplot – 25th percentile
> Top edge of the boxplot – 75th percentile
> Outliers: Datapoints greater than $75\text{th quartile} + \left(1.5 * \text{interquartile range}\right)$ or less than $25\text{th quartile} - \left(1.5 * \text{interquartile range}\right)$
> Whiskers: Extreme datapoints other than outliers ($\pm 2.7 * \text{standard deviation}$ for normally distributed data).

We have also computed the magnitude-squared coherence between the stimulus and response time domain signals in all the groups at the stimulus frequencies in MATLAB (*Figure 3—figure supplement 2*). Magnitude squared coherence is a function of power spectral densities of stimulus, response, and their cross-power spectral density, assuming values range from 0 to 1. A value of 1 indicates that the stimulus and response time domain signals are well matched at that frequency.

## Statistical tests

Because the data were not normally distributed, we used non-parametric tests to statistically compare the head-compensation error between the different treatment groups in all cases. As the light and dark trials were conducted on the same animal (in *Figure 2C*), we used a paired Wilcoxon's signed-rank test to compare the data. We compared the *control*, *flagella-clipped*, and *flagella-reattached* groups using a Kruskal-Wallis test as the groups were independent. To further identify which groups were statistically different, the Kruskal-Wallis test was followed by a post hoc comparison with a Nemenyi test. The unpaired data (in *Figure 4C*) were compared using a Wilcoxon rank-sum test, a non-parametric test used for statistically comparing two independent groups.

## Free-flight assay

To address whether restriction of head movements in moths impact their free-flight performance, we conducted the following experiments.

## Method details

Moths were divided into three groups including *control*, *sham,* and *head-restricted* moths. Initially, all the groups were cold-anesthetized by placing them in –20°C for approximately 8–12 min. *Control* moths did not undergo any head restriction treatment, but went through the same anesthesia procedures as the other two groups. In the experimental *head-restricted* group, we partially removed the scales in the dorsal neck-prothorax region using a paintbrush and put tiny drops of molten dental wax at the neck using a heated wire. The wax solidified to make a bridge between prothorax and the posterior head, restraining any relative motion between head and thorax. In the *sham* group, we partially descaled the dorsal neck-prothorax region and put a similar amount of molten wax on another part of the prothorax, thus allowing the head to freely move relative to the thorax. All the moths were allowed to recover for 1 hr after the treatment.

The hawkmoths were next released inside a transparent acrylic flight chamber (*Figure 6—figure supplement 2A*) and their flight performance were observed and filmed both before (as internal control) and after treatment for each of the *control*, *sham,* and *head-restricted* groups. These

experiments were conducted in a large greenhouse in the evening around dusk during their naturally active periods. Before the experiments commenced, moths with obvious flight defects were discarded. Moths with no obvious flight defects underwent the three treatment conditions and their flight performance was filmed. Two tripod-mounted monochrome cameras (Pointgrey BFS-U3-13Y3M) were placed orthogonally and synced together to film the behavior simultaneously through transparent walls of the flight chamber (*Figure 6—figure supplement 2A*). A low-power bulb (Philips Deco Yellow 15 watts) hung approximately 15 cm above the ceiling of the flight chamber provided nearly uniform, low-intensity illumination inside the arena. The light intensity, as measured by a light meter (center 337; range: 0.01–40,000 lux), was ~1.56 lux at the center of the arena's base, the area around which the moth was released. The moth was stimulated with air-puff to initiate flight. The behavior was filmed at a resolution of 1280 × 1024 pixels at 30 frames per second with an exposure time of 33 ms.

In a separate batch of moths, we tested the efficacy of the wax treatment by restricting the insect head movements using the previously described tethered assay for roll stabilization (*Figure 1A*). Using this assay, we ascertained that head movements relative to the thorax were completely abolished in moths with waxed neck joints (*Figure 6—figure supplement 2B-D*).

## Quantification and statistical analysis

### Analysis

The duration of a flight bout was defined as the time spent between post-warmup take-off and termination of wing flapping upon landing. The take-off time was noted when the legs left the ground, whereas the landing time was noted when the moth finally touched down and ceased flapping its wings. In some cases, the partially airborne moth skidded on the ground or crawled with forelegs touching the side walls even while the wings were flapping at high amplitude (*Video 3*). These moths were not considered flying as long as their legs touched the ground, and thus time spent in any intermediate skidding and crawling was subtracted from the total duration of a flight bout. Collisions with the side walls were observed and noted. Skidding and crawling were not considered as collisions, except the initial contact with the wall. Collision frequency was calculated as the total number of collisions with any side wall divided by the flight bout duration.

Although these videos were filmed at 30 fps, each video frame encompassed a large volume of the flight chamber (1 m × 1 m × 0.8 m), and hence the moths occupied a very small fraction of image. Moreover, the low-light conditions meant that some blur was added to the image during rapid movements. This made it difficult for us to determine the precise instant at which the legs were off the ground (to determine flight onset) or when they touched the ground (to determine the flight cessation). Thus, when measuring the time duration of flight, we have rounded off the time to the nearest second resulting in some discretization in the estimation of collision/s (for instance, 5/8 points in *Figure 6B* occur at 1 collision/s).

### Statistical analysis

Because the groups were unpaired and did not have a Gaussian distribution, we statistically compared flight bout duration and collision frequency between the *control*, *sham*, and *head-restricted* moths using a non-parametric Kruskal-Wallis test, followed by a post hoc Nemenyi test (*Figure 6*). As the before and after treatments within each of *control*, *sham*, and *head-restricted* groups were conducted on the same animal (in *Figure 6—figure supplement 1*), we used a paired Wilcoxon's signed-rank test to compare the data.

## Acknowledgements

We thank M Kemparaju and Allan Francis Joy for maintaining the moth culture, Shivansh Dave for his help with setting up the stepper motor, free-flight arena set-up, camera synchronization and free-flight pilot experiments, and the mechanical and electronics workshops at NCBS for help with the apparatus. Funding for this study was provided by grants from Air Force Office of Scientific Research (AFOSR) # FA2386-11-1-4057 and # FA9550-16-1-0155, and National Centre for Biological Sciences (Tata Institute of Fundamental Research) to SPS. We acknowledge support of the Department of Atomic Energy, Government of India, under project no. 12 R&D-TFR-5.04–0800 and NCBS core computational facilities, supported under 12 R&D-TFR-5.04–0900.

## Additional information

### Funding

| Funder | Grant reference number | Author |
|---|---|---|
| Air Force Office of Scientific Research | FA2386-11-1-4057 | Sanjay P Sane |
| Air Force Office of Scientific Research | FA9550-16-1-0155 | Sanjay P Sane |
| Department of Atomic Energy, Government of India | 12-R&D-TFR-5.04-0800 | Sanjay P Sane |
| National Centre for Biological Sciences | 12-R&D-TFR-5.04-0900 | Sanjay P Sane |

The funders had no role in study design, data collection and interpretation, or the decision to submit the work for publication.

### Author contributions

Payel Chatterjee, Conceptualization, Data curation, Formal analysis, Investigation, Methodology, Software, Validation, Visualization, Writing – original draft, Writing – review and editing; Agnish Dev Prusty, Data curation, Investigation, Methodology, Software, Validation, Visualization, Writing – original draft, Writing – review and editing; Umesh Mohan, Conceptualization, Formal analysis, Investigation, Methodology, Software, Validation, Visualization, Writing – original draft, Writing – review and editing; Sanjay P Sane, Conceptualization, Funding acquisition, Investigation, Project administration, Resources, Supervision, Validation, Visualization, Writing – original draft, Writing – review and editing

### Author ORCIDs

Payel Chatterjee http://orcid.org/0000-0002-7019-6702
Agnish Dev Prusty http://orcid.org/0000-0002-4890-9163
Umesh Mohan http://orcid.org/0000-0002-1992-0558
Sanjay P Sane http://orcid.org/0000-0002-8274-1181

### Decision letter and Author response

Decision letter https://doi.org/10.7554/eLife.78410.sa1
Author response https://doi.org/10.7554/eLife.78410.sa2

## Additional files

### Supplementary files

• Supplementary file 1. (1a) List of interquartile range of compensation error. (1b) List of interquartile ranges of coherence. (1c) Details of the statistical tests used in various experiments in this paper.
• MDAR checklist

### Data availability

All data related to this paper (both raw and processed) are available on the following link: https://data.mendeley.com/datasets/2trxj9gwsw/draft?a=89356c26-e581-40c6-935d-c1d4a0401074.

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

## Appendix 1

We have estimated power that is the probability of detecting a difference between the groups with a significance level of 0.05 across a range of sample sizes for all the comparisons which yielded significant difference. We have used the mean and standard deviation of each dataset for generating normal distributions. We have then used the relevant tests that is Wilcoxon signed-rank, Wilcoxon rank-sum, and Kruskal-Wallis tests for the power analysis. For each sample size, the loop was repeated 1000 times. In all the plots, the power of 0.8 is marked in dotted line. Our sample size in all cases is greater than or close to the power of 0.8.

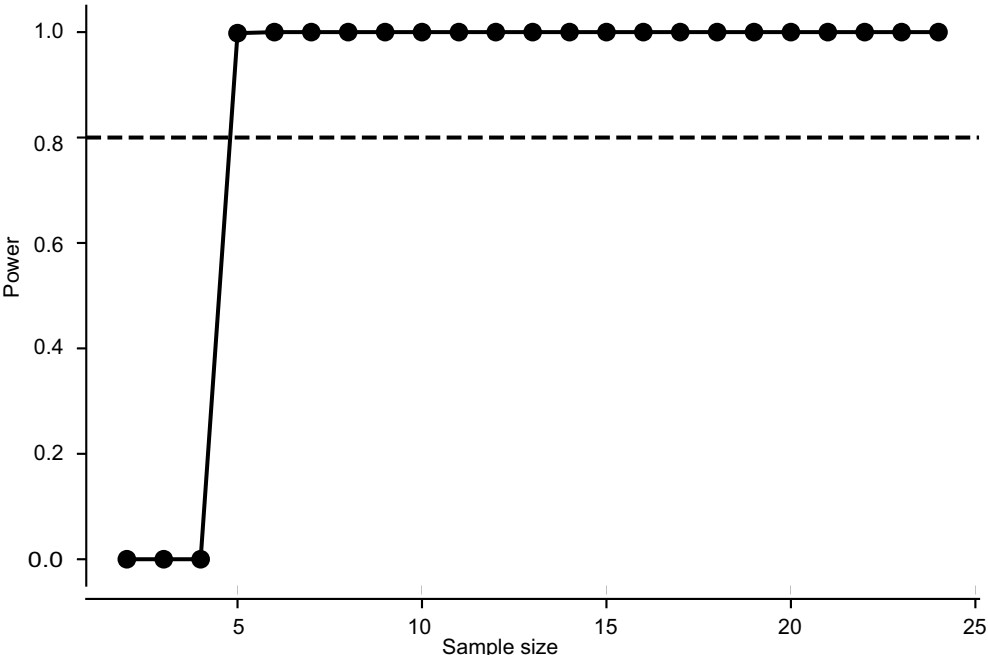

**Appendix 1—figure 1.** Sample size estimation for twilight vs. dark condition in *Figure 2*. The minimum sample size required for having a power of 0.8 is 5. The sample size in this case: n=8 (twilight) and n=8 (dark) (frequency: 2 Hz imposed roll).

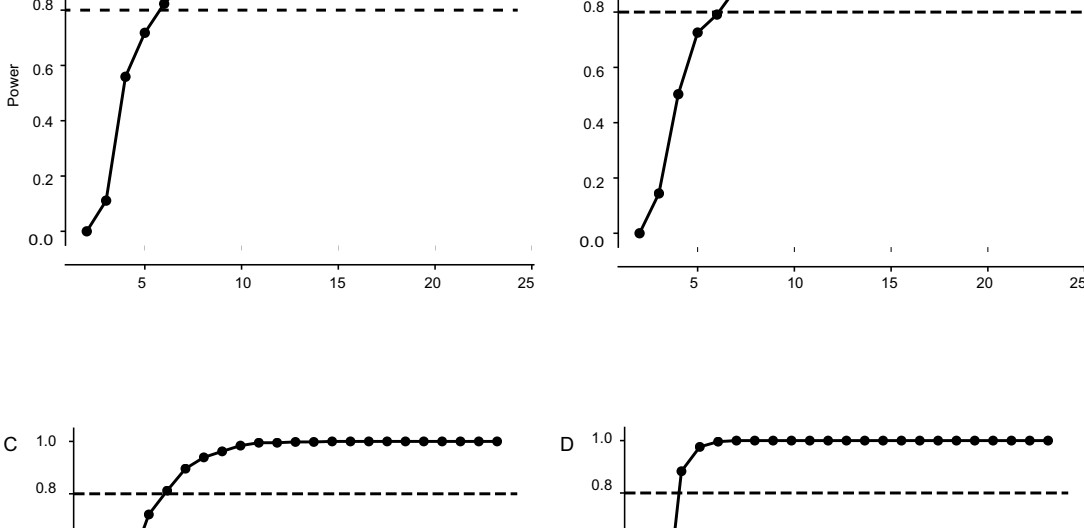

**Appendix 1—figure 2.** Sample size estimation for control, flagella-clipped, and flagella-reattached moths in *Figure 3*. (**A**) The minimum sample size required for having a power of 0.8 is 6. The sample size in this case: control: n=8, flagella-clipped: n=7, flagella-reattached: n=8 (frequency: 2 Hz imposed roll, light level: twilight). (**B**) The minimum sample size required for having a power of 0.8 is 6. The sample size in this case: control: n=8, flagella-clipped: n=7, flagella-reattached: n=8 (frequency: 2 Hz imposed roll, light level: dark). (**C**) The minimum sample size required for having a power of 0.8 is 6. The sample size in this case: control: n=8, flagella-clipped: n=7, flagella-reattached: n=8 (frequency: 6 Hz imposed roll, light level: twilight). (**D**) The minimum sample size required for having a power of 0.8 is 4. The sample size in this case: control: n=8, flagella-clipped: n=7, flagella-reattached: n=8 (frequency: 6 Hz imposed roll, light level: dark).

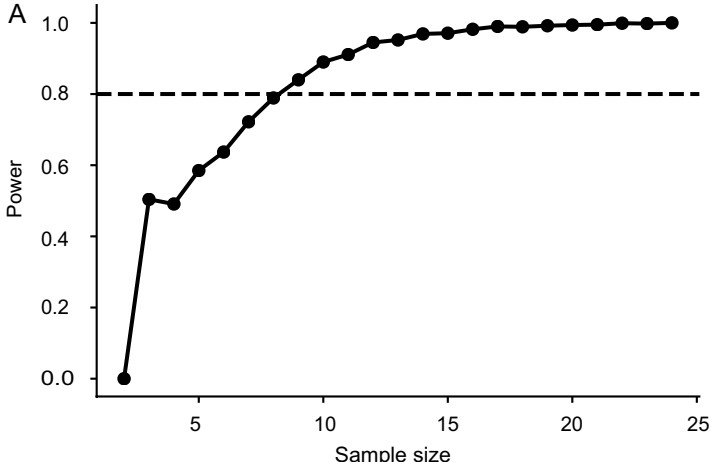

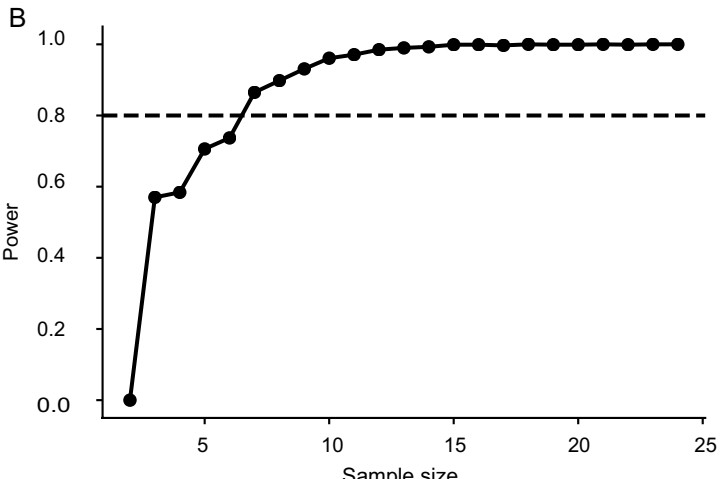

**Appendix 1—figure 3.** Sample size estimation for sham and Johnston's organ-glued moths in *Figure 4*. The minimum sample size required for having a power of 0.8 is 9. The sample size in this case: sham: n=10, Johnston's organ-glued: n=7 (frequency: 6 Hz imposed roll, light level: twilight). (**B**) The minimum sample size required for having a power of 0.8 is 7. The sample size in this case: sham: n=9, Johnston's organ-glued: n=8 (frequency: 6 Hz imposed roll, light level: dark).

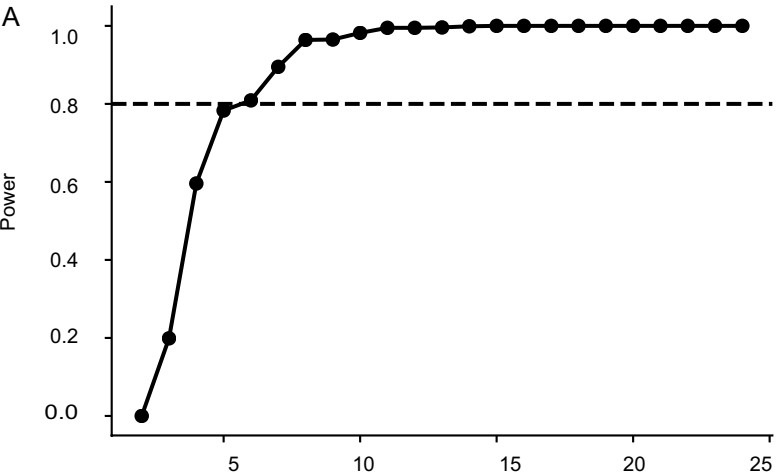

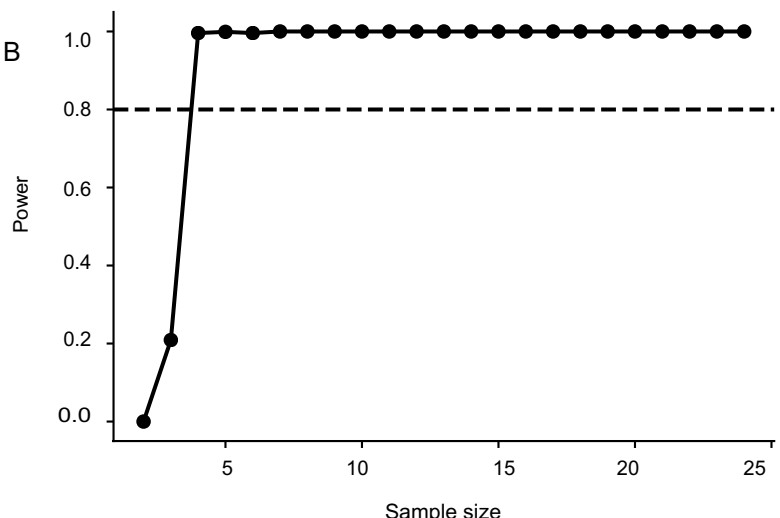

**Appendix 1—figure 4.** Sample size estimation for control, sham, and head-restricted moths in *Figure 6*. The minimum sample size required for having a power of 0.8 is 5. The sample size in this case: control: n=11, sham: n=12, head-restricted moths: n=0.8 (flight bout duration). (**B**) The minimum sample size required for having a power of 0.8 is 4. The sample size in this case: control: n=11, sham: n=12, head-restricted moths: n=.8 (collision frequency).

