## [Editor Report]

This paper will be of interest to neuroscientists who study navigation and multisensory integration. In it, the authors use several manipulations to convincingly show that hawkmoths use mechanosensory feedback from their antennae to stabilize their head when their body rotates quickly or when they have little visual input. The results are consistent with the hypothesis that control of head angle in insects that lack halteres results from a multimodal feedback loop that integrates visual and antennal mechanosensory feedback. This advances our understanding of how such stabilizing reflexes work beyond Dipteran flies, where much prior work has focused.

---

## [Decision Letter]

**Decision letter after peer review:**

Thank you for submitting your article "Integration of visual and antennal mechanosensory feedback during head stabilization in hawkmoths" for consideration by *eLife*. Your article has been reviewed by 3 peer reviewers, one of whom is a member of our Board of Reviewing Editors, and the evaluation has been overseen by Ronald Calabrese as the Senior Editor. The following individual involved in review of your submission has agreed to reveal their identity: Marie Dacke (Reviewer #3).

Essential revisions:

The reviewers agreed that there were four essential revisions the authors should make.

1) This study focuses on how antennal information is integrated with visual information to stabilize head roll. However, the authors do not provide much information about what the visual environment was for these studies. This should be rectified.

2) The authors should provide some concrete (quantitative?) predictions made by the feedback controller model and compare those to the data.

3) The single scalar error metric should be better explained, and more effort should be made to explain why/how it is superior to examining the gain and phase separately (or jointly). (Or the authors could adjust their metrics for error.)

4) The detailed reviews note a few cases in which figure clarity could be improved, and this would benefit future readers.

*Reviewer #1 (Recommendations for the authors):*

Chatterjee et al. present a convincing set of findings showing that hawkmoths use their antennae to sense and (partially) correct for head rolls. The manipulations, and especially the re-attached antennal segments, were especially convincing. I noted a few places where I thought the paper could be more complete, noted below.

1) The visual feedback was important to this behavior, but I didn't see any description of what the visual environment was about the moth during these experiments. This seems important to describe. This relates also to line 372 of the discussion, where the authors discuss the "3d distribution of objects"? I didn't follow what this sentence was referring to, and I generally did not follow the discussion on this line through line 378.

2) Figure 5. I would love to see predictions of this model for the various cases here: visual and antennal feedback, or one eliminated, at 2 Hz and 6 Hz… With the simplest possible parameters and assumptions about linearity, I guess, which give the qualitative responses seen here. It might be nice to compare the gain-phase plots to data under some of these different parameters…

3) I was a little puzzled by the error metric, \epsilon. If \epsilon ("compensation error") is a scalar (Cartesian) deviation in the polar gain plot from (1,180d), then I don't really understand why it's constrained to be in the range [0,2]. Couldn't the gain be 4 and the phase be 180 d? This single number also seems to have some strange properties – it makes the most sense when epsilon is small. But in Figure 2C, the phase differences between conditions in 6 Hz case don't show up with this metric – and maybe didn't matter in that case. But one could imagine other cases where one could be led astray. Is there any reason not to just have a gain and a phase and do statistics on each separately for the different conditions? This would provide more information – for example, emphasizing that some manipulations changed phases, but not gains, which is visible in the current data, but not emphasized by statistical tests.

*Reviewer #2 (Recommendations for the authors):*

A few changes to the text and figures would improve clarity and ease of reading.

1) The conceptual model shown in Figure 5 is helpful, but it would be nice to actually show some simulations with this model and compare them to the experimental data.

2) The authors might provide further quantitative descriptions of how flight trajectories change after head stabilization in Figure 6.

*Reviewer #3 (Recommendations for the authors):*

1) There are a few points regarding the methods that needs to be described more in detail prior to publication. My main concern is about the visual environment of the moth in the experimental set-ups. Were visual structures for head stabilization provide by the general clutter of the set-up or was the environments more controlled? For minor comments, see below.

2) Could the authors please explain why they did not include the flagella-clipped moths also in the free flight experiments. This would have provided a valuable dataset for comparison with the head-stabilized moths as both groups could, according to the conclusion of this study, be expected to be similarly affected.

---

## [Author Response]

Essential revisions:The reviewers agreed that there were four essential revisions the authors should make.1) This study focuses on how antennal information is integrated with visual information to stabilize head roll. However, the authors do not provide much information about what the visual environment was for these studies. This should be rectified.

Please refer to our response to Comment #1 by Reviewer #1, and Comment #1 by Reviewer #3.

2) The authors should provide some concrete (quantitative?) predictions made by the feedback controller model and compare those to the data.

Please refer to our response to Comment #2 by Reviewer #1, and Comment #1 by Reviewer#2.

3) The single scalar error metric should be better explained, and more effort should be made to explain why/how it is superior to examining the gain and phase separately (or jointly). (Or the authors could adjust their metrics for error.)

Please refer to our response to Comment #3 by Reviewer #1.

4) The detailed reviews note a few cases in which figure clarity could be improved, and this would benefit future readers.

We have incorporated all the figure-related changes suggested by the reviewers.

Reviewer #1 (Recommendations for the authors):Chatterjee et al. present a convincing set of findings showing that hawkmoths use their antennae to sense and (partially) correct for head rolls. The manipulations, and especially the re-attached antennal segments, were especially convincing. I noted a few places where I thought the paper could be more complete, noted below.1) The visual feedback was important to this behavior, but I didn't see any description of what the visual environment was about the moth during these experiments. This seems important to describe. This relates also to line 372 of the discussion, where the authors discuss the "3d distribution of objects"? I didn't follow what this sentence was referring to, and I generally did not follow the discussion on this line through line 378.

Thanks for this feedback. Yes, we agree with the reviewers that the wording “3D distribution of objects” is unclear, so we have now elaborated on this description in the methods section. We consciously opted to work in the standard lab environment, instead of using a 2D display screen (e.g. a chequerboard pattern printed on paper, or a video display) because:

a) Previous studies have highlighted the importance of 3D cues for flight. In our experimental design, we therefore wanted to retain 3D / depth cues in the visual environment presented to the moths. We are mindful that for pure roll motion, depth/parallax cues are minimal. However, in these experiments the head was not glued to the thorax, and hence we could not rule them out.

b)More importantly: in this paper, we were careful to ensure that each experimental treatment was always compared with its corresponding control. This means that we focused only on the differences between the control and experimental scenarios which ensures that our results and conclusions should be robust for any arbitrary visual environment. Again, for this, the regular laboratory environment was sufficient.

Figure 1—figure supplement 1 shows images of (a) the moth’s visual field during these experiments, (b) of the apparatus to which the moth was tethered and oscillated during the experiments, and (c) a layout of the overall room layout and its dimensions.

2) Figure 5. I would love to see predictions of this model for the various cases here: visual and antennal feedback, or one eliminated, at 2 Hz and 6 Hz… With the simplest possible parameters and assumptions about linearity, I guess, which give the qualitative responses seen here. It might be nice to compare the gain-phase plots to data under some of these different parameters…

The model in Figure 5 is a conceptual and not yet a formal control model. It was constructed based on the experimental data, but quantitative predictions will only be possible after the model has a proper formal (mathematical) structure. We had earlier on tried to develop a model based on linearity assumptions, but soon realized that we lacked key inputs that were required to simulate the multimodal head stabilization loop. The key missing inputs for a formal model include neck muscle / joint mechanics, estimates of delay and damping coefficients, dynamics of the visual and antennal mechanosensory systems, and the nature of the operators (i.e. linear or non-linear). We also have no information on the neural controller block, which integrates visual and antennal mechanosensory inputs to generate a motor output. Moreover, there are strong indications that such integration could be non-linear (e.g. Huston and Krapp, 2009, Haag et al., 2010), and thus assuming linearity at this stage may not be correct.

Nevertheless, the conceptual model presented here can make specific testable predictions, as described below:

Any process that is modelled as a negative feedback loop has inherent oscillatory dynamics that depends on the latencies of the feedback and actuation. If these latencies are well-matched and fast, then the system’s output will undergo small-amplitude oscillations about the set point. However, if we increase the latencies of the feedback by modifying the sensory feedback, then we expect that the system will find it harder to maintain a stable state and the fluctuations will be of larger amplitude, as the system strays more from its set point.

Because head stabilization is modelled here as a multimodal (antennal mechanosensory and visual) negative feedback loop, we expected that head position should also be in a state of dynamic equilibrium. If its dynamics depends on feedback from both systems, then altering the antennal and visual feedback should also cause alterations in this dynamic equilibrium. From this conceptual model, we predict that this dynamic equilibrium will manifest as small-amplitude oscillations which will change their properties if we manipulate each feedback.

We observed exactly such small-amplitude oscillations, which we then experimentally tested by manipulating the feedback. These data have been submitted as a separate paper (uploaded to https://www.biorxiv.org/content/10.1101/2022.04.24.489321v1). We have now added a new section in the discussion in which we mention these oscillations as a predicted outcome of the negative feedback loop, with a citation to this BioRxiv paper, which is in review.

3) I was a little puzzled by the error metric, \epsilon. If \epsilon ("compensation error") is a scalar (Cartesian) deviation in the polar gain plot from (1,180d), then I don't really understand why it's constrained to be in the range [0,2]. Couldn't the gain be 4 and the phase be 180 d? This single number also seems to have some strange properties – it makes the most sense when epsilon is small. But in Figure 2C, the phase differences between conditions in 6 Hz case don't show up with this metric – and maybe didn't matter in that case. But one could imagine other cases where one could be led astray. Is there any reason not to just have a gain and a phase and do statistics on each separately for the different conditions? This would provide more information – for example, emphasizing that some manipulations changed phases, but not gains, which is visible in the current data, but not emphasized by statistical tests.

Thank you for this comment. In a nutshell: performing statistics separately on gain and phase would be both misleading and incorrect. To illustrate this point, consider the following hypothetical dataset shown in Author response image 1.

**Author response image 1. sa2fig1:** 

If gain and phase are separate errors, then the mean of the above dataset would be gain = 0.5, phase = 0 (blue “x” in plot) which is incorrect. The actual mean is gain = 0.3898, phase = 0 (green filled circle in plot). This example illustrates why it is misleading to interpret or statistically analyse gain and phase separately.To remedy this, following the method similar to that outlined in Sponberg et al. (2015), Dahake et al. (2018), and Windsor and Taylor (2017), we constructed a tracking error metric (in our case, the compensation error) which combines gain and phase in a manner that readily enables statistical comparisons. However, this metric also does not by itself present a full picture. We need all three quantities: gain, phase, and compensation error, to obtain an unambiguous picture. Hence, in the manuscript, we have provided all three quantities to the readers.

In general, it is always challenging to find an optimal metric that can describe the nuances of a complex behaviour using a single number. We explain below why the error metric as outlined here is a reasonable way of presenting these results. The above query is broken down into subparts, each answered independently.

a) why is the compensation error constrained to be in the range [0,2]. Couldn't the gain be 4 and the phase be 180 d?

In principle, the compensation error can exceed 2 if the gain exceeds 1 (i.e. the head moves much more than the stimulus). Although theoretically possible, this was never observed either in our system, or in any other study of head stabilization in the literature. For this reason, the Y-axis in the compensation error plots range from 0-2. We have clarified this in the methods section. The added portion now reads, “In this representation, the compensation error (ɛ) can range from 0 to 2 (for gain ≤ 1) with an ɛ of 0 representing perfect head stabilization and corresponds to (r = 1, <inline-graphic mimetype="image" mime-subtype="png" xlink:href="media/image1.png" /> = 180^º^) in the polarscatter plots (Figure 1E).”

To aid the intuition further, we have plotted Figure 2A with colour gradient of compensation error to better relate gain-phase values with compensation error, and added this in the Supplementary section (Figure 2 —figure supplement 2)

b) But in Figure 2C, the phase differences between conditions in 6 Hz case don't show up with this metric – and maybe didn't matter in that case. But one could imagine other cases where one could be led astray.

As stated above, the compensation error metric is, by itself, incomplete. To get the full picture, we need to combine it with phase and gain values which were provided in the polarscatter plots and supplementary information.

Reviewer #2 (Recommendations for the authors):A few changes to the text and figures would improve clarity and ease of reading.1) The conceptual model shown in Figure 5 is helpful, but it would be nice to actually show some simulations with this model and compare them to the experimental data.

Thanks much for this comment. As explained in our response to the comment #2 for Reviewer#1, the effort to quantitatively model or simulate this system was premature at this stage because we lacked some key mechanical and dynamical inputs, such as the mechanics of neck muscles, values of various latencies, and the nature of various transformation functions, which we do not yet have. However, we certainly plan to provide a more formal treatment of the model in the near future.

Nevertheless, our experimental data are consistent with the specific predictions from the multimodal negative-feedback model presented here, as explained in our response to Reviewer#1. The small-amplitude head oscillations (head wobble) observed in our system may be arise from to the inherent dynamics of such a feedback loop.

2) The authors might provide further quantitative descriptions of how flight trajectories change after head stabilization in Figure 6.

This comment was not clear. Perhaps, Reviewer #2 meant that we should provide the flight trajectories after “head-restriction" rather than “head-stabilization”. In any event, this experiment (which recapitulates the free flight experiments in Sane et al. 2007) was not set up to measure the flight trajectories, which would require two synchronized and calibrated high-speed cameras which were not used here, especially because we were filming over large time periods. Instead, our focus was on quantifying the gross flight defects in a freely flying moth in the style of Sane et al., 2007.

Reviewer #3 (Recommendations for the authors):1) There are a few points regarding the methods that needs to be described more in detail prior to publication. My main concern is about the visual environment of the moth in the experimental set-ups. Were visual structures for head stabilization provide by the general clutter of the set-up or was the environments more controlled?

Thank you for this feedback. This was also the question in comment #1 of R#1. To summarize that response: the moths were filmed within the laboratory environment.

The surrounding visual features were kept constant across all the treatments. Because all our conclusions were based on comparisons with the corresponding controls, we are confident that these results should hold true in any arbitrary visual environment. We have now included additional text in the Methods and supplementary figures to explain the visual structure of the moths

2) Could the authors please explain why they did not include the flagella-clipped moths also in the free flight experiments. This would have provided a valuable dataset for comparison with the head-stabilized moths as both groups could, according to the conclusion of this study, be expected to be similarly affected.

The experiment suggested by the reviewer (flagella-clipped moths in free-flight experiments) were described by Sane et al., 2007 in the hawkmoth *Manduca sexta*, and are consistent with our observations in *Daphnis nerii*. That paper is cited here. Here too, we assume that the referee meant “head-restricted” rather than “head- stabilized” moths. Head restriction already causes a severe flight defect (as shown here), as does flagella-clipping (as shown in Sane et al. 2007). Thus, combining these two strong effects, may not be particularly informative, as it is difficult to parse which of these manipulations causes the catastrophic failure